# MDA5 disease variant M854K prevents ATP-dependent structural discrimination of viral and cellular RNA

Qin Yu [1,2,3,4], Alba Herrero del Valle [1,2,4], Rahul Singh [1,2,4] & Yorgo Modis [1,2✉]

Our innate immune responses to viral RNA are vital defenses. Long cytosolic double-stranded RNA (dsRNA) is recognized by MDA5. The ATPase activity of MDA5 contributes to its dsRNA binding selectivity. Mutations that reduce RNA selectivity can cause autoinflammatory disease. Here, we show how the disease-associated MDA5 variant M854K perturbs MDA5-dsRNA recognition. M854K MDA5 constitutively activates interferon signaling in the absence of exogenous RNA. M854K MDA5 lacks ATPase activity and binds more stably to synthetic Alu:Alu dsRNA. CryoEM structures of MDA5-dsRNA filaments at different stages of ATP hydrolysis show that the K854 sidechain forms polar bonds that constrain the conformation of MDA5 subdomains, disrupting key steps in the ATPase cycle-RNA footprint expansion and helical twist modulation. The M854K mutation inhibits ATP-dependent RNA proofreading via an allosteric mechanism, allowing MDA5 to form signaling complexes on endogenous RNAs. This work provides insights on how MDA5 recognizes dsRNA in health and disease.

[1] Molecular Immunity Unit, Department of Medicine, University of Cambridge, MRC Laboratory of Molecular Biology, Francis Crick Avenue, Cambridge Biomedical Campus, Cambridge CB2 0QH, UK. [2] Cambridge Institute of Therapeutic Immunology & Infectious Disease (CITIID), University of Cambridge School of Clinical Medicine, Cambridge CB2 0AW, UK. [3] Present address: Institute of Molecular Biology and Biophysics, Department of Biology, ETH Zürich, Zürich, Switzerland. [4] These authors contributed equally: Qin Yu, Alba Herrero del Valle, Rahul Singh. ✉email: ymodis@mrc-lmb.cam.ac.uk

Viruses deliver or generate RNA in the cytosol of the host cell to have their genes expressed and replicated by cellular machinery. Recognition of cytosolic viral double-stranded RNA (dsRNA) is one of the most important and conserved mechanisms for sensing infection. dsRNA is recognized in the cytosol by the innate immune receptors RIG-I, MDA5, and LGP2. RIG-I binds dsRNA blunt ends with di- or triphosphorylated 5′-ends[1–3]. MDA5 cooperatively assembles into helical filaments on uninterrupted RNA duplexes longer than a few hundred base pairs (bp)[1,2,4–6]. LGP2 promotes and stabilizes MDA5-RNA complex formation[7,8]. RIG-I, MDA5, and LGP2 bind dsRNA with a modified DExD/H-box helicase core and a C-terminal domain (CTD)[9–14]. The helicase consists of two RecA-like domains, Hel1 and Hel2, and an insert domain, Hel2i, all of which form contacts with phosphate and ribose moieties of both RNA strands. The helicase and CTD, linked by a pair of α-helices referred to as the pincer domain, form a ring around the RNA. Upon binding RNA, RIG-I-RNA oligomers and MDA5-RNA filaments nucleate the assembly of MAVS into microfibrils[15,16], which then recruit proteins from the TRAF and TRIM families to activate potent type I interferon and NF-κB inflammatory responses[1,2,15].

MDA5 is the primary innate immune sensor for many viruses[1,2], including SARS-CoV-2[17,18], but viral dsRNA can be difficult to distinguish from endogenous dsRNAs. Retrotransposon transcripts, including inverted Alu repeats within mRNA and bidirectionally transcribed LINE-1 elements can anneal into RNA duplexes that are sensed by MDA5 as viral dsRNA[19–22]. Retrotransposons constitute one-third of the human genome. A-to-I deamination by ADAR1 destabilizes Alu:Alu duplexes sufficiently to prevent MDA5 filament formation[19,20]. Nevertheless, efficient proofreading and fine-tuned signal transduction are necessary for cells to respond appropriately to infection without triggering autoimmune responses. Cryogenic electron microscopy (cryoEM) structures of MDA5-dsRNA filaments at different stages of the ATPase cycle—bound to ATP, transition-state analog ADP-AlF$_4$, or no nucleotide—showed that ATP hydrolysis by MDA5 is coupled to conformational changes in MDA5-dsRNA filaments[14]. We proposed the ATPase cycle performs a mechanical proofreading function by testing the interactions with the bound RNA, promoting dissociation of MDA5 from loosely-bound endogenous RNAs while allowing it to remain bound to viral RNAs long enough to activate signaling[14].

Mutations that alter the parameters governing MDA5-dsRNA filament dynamics can cause inappropriate activation of the antiviral innate response. MDA5 gain-of-function mutations and ADAR1 loss-of-function can cause PKR-mediated translational shutdown and severe autoinflammatory disorders[19,20,23,24]. Approximately 30 MDA5 point mutations have been linked to severe autoinflammatory disorders in heterozygous human patients, including Aicardi–Goutières syndrome (AGS), Singleton–Merten syndrome (SMS), spastic-dystonic syndrome, and neuroregression[25–27]. Pathogenic MDA5 mutations are associated with increased expression of interferon-stimulated genes in patients[27] and increased IFN-β signaling in cultured cells, particularly without stimulation with exogenous dsRNA[26]. MDA5 mutations are known or predicted to be a pathogenic map to the RNA and ATP binding sites, with a few exceptions (Supplementary Fig. 1a). Mutations in the ATP binding pocket inhibit ATP hydrolysis without affecting filament formation, whereas those at the RNA interface increase the affinity or avidity of MDA5 for RNA[26]. In either case, the consequence is to stabilize MDA5-dsRNA filaments and reduce the proofreading activity of MDA5, leading to constitutive interferon signaling from binding to endogenous dsRNAs, along with an increased signaling

response to viral dsRNA[14,19]. A few pathogenic MDA5 mutations map outside the ATP and RNA binding sites. The M854K variant (encoded by missense mutation 2561 T > A in the *IFIH1* gene), found in patients from two unrelated families, is associated with symptoms of AGS and SMS caused by elevated type I interferon signaling[27–29]. Residue 854 is located on the first pincer helix, distant from the ATP and RNA binding sites. Therefore, the molecular mechanism of disease pathogenesis from the M854K mutation remains unknown.

Here, we examine the consequences of the M854K disease variant on the structure and activities of MDA5. We show that the M854K mutation abrogates ATPase activity, stabilizes MDA5-dsRNA complexes, increases IFN-β transcription, and dampens the cooperativity of signal transduction. CryoEM structures of WT and M854K MDA5-dsRNA filaments at different stages of ATP hydrolysis, determined at resolutions of up to 2.8 Å, reveal in detail how the M854K mutation constrains conformational changes in MDA5 necessary for ATP hydrolysis and ATPase-dependent dissociation from endogenous RNAs, ultimately leading to constitutive signaling activation.

## Results

**MDA5 disease variant M854K confers constitutive IFN-β signaling activity.** Previously studied pathogenic MDA5 variants stabilize MDA5-dsRNA filaments by reducing the ATPase activity or increasing the RNA binding affinity of MDA5, leading to increased interferon signaling[26]. To determine the functional consequences of the M854K mutation, we measured its activity in a cell signaling assay. We compared the M854K variant to WT MDA5 and to variants with point mutations at the filament-forming interface engineered to increase or decrease signaling activity (Fig. 1)[14]. Expression plasmids were transfected into HEK293T cells together with plasmids encoding firefly luciferase under the control of the IFN-β promoter and *Renilla* luciferase under a constitutive promoter. Cells were subsequently transfected with poly(I:C) RNA, a dsRNA analog, to induce MDA5 signaling. IFN-β-dependent gene expression was measured as the ratio of firefly to *Renilla* luciferase luminescence[14]. The expression level of each MDA5 variant was assessed by Western blotting (Fig. 1). The luciferase activity of the M854K mutant was 19-fold higher than WT without poly(I:C) stimulation, and 20–40% higher than WT with poly(I:C) stimulation (Fig. 1a). By comparison, the activity of variant H871A/E875A, shown previously to be hyperactive[14], was ninefold higher than WT without poly(I:C) stimulation, and 10–30% higher than WT with poly(I:C) stimulation (Fig. 1a). Deletion of the C-terminal 12 residues (ΔC12), which forms MDA5-dsRNA aggregates instead of helical filaments[14], resulted in no signaling activity without poly(I:C) stimulation and 50% of WT activity with stimulation. The D848K/F849A/R850E mutations, which completely inhibit filament formation but not ATPase activity[14], abolished signaling with and without poly(I:C) stimulation (Fig. 1a). We conclude that the primary effect of the M854K mutation is to confer constitutive signaling activity in the absence of exogenous RNA. This is consistent with the reported upregulation of type I interferon signaling in a patient harboring the M854K mutation, in the absence of viral infection[28]. The effect of M854K on signaling is distinct from that of gain- or loss-of-function mutations at the filament interface, but similar to that of disease mutations in or near the ATP binding site, such as R337G and R779C/H, or RNA binding interface, such as D393V and G495R[19,26,30].

MDA5 recognizes both the structure and length of its dsRNA ligands. The length specificity stems from the RNA binding cooperativity of MDA5, encoded by its filament-forming interfaces[4,5]. The structural selectivity of MDA5 arises in part

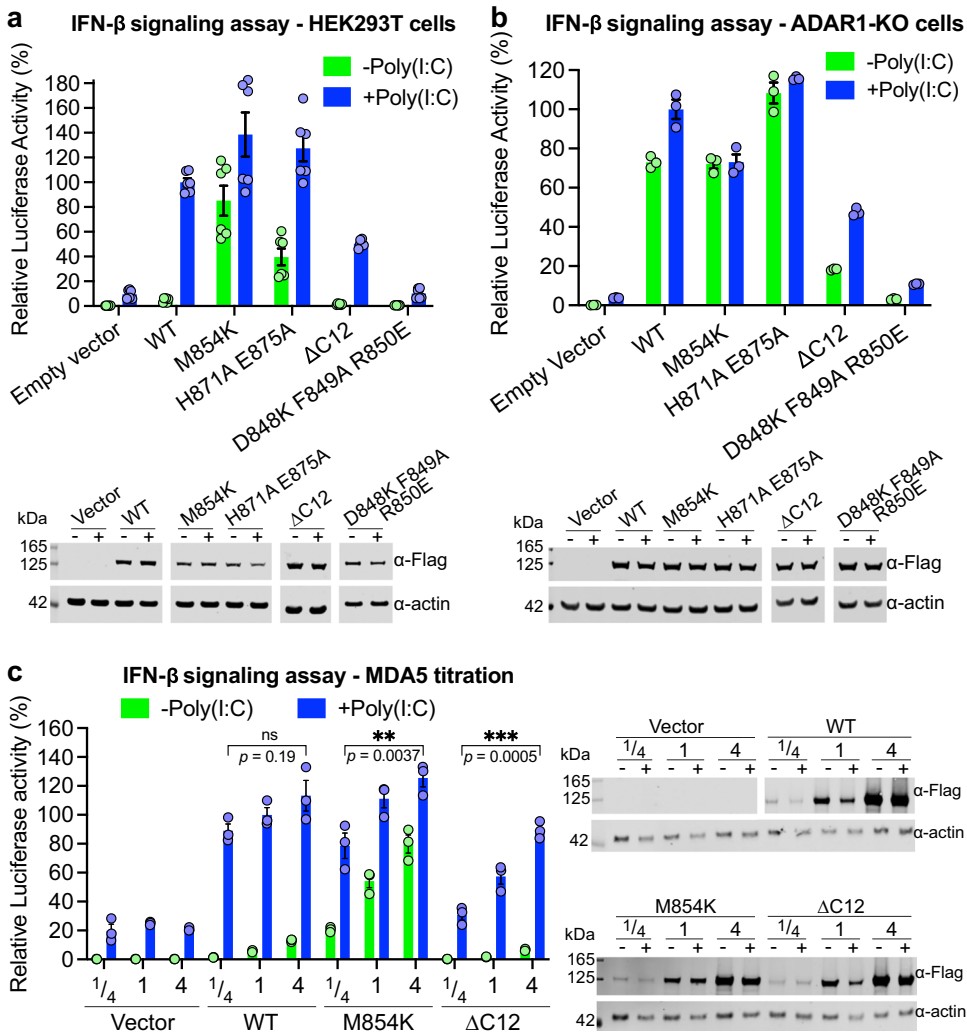

**Fig. 1 MDA5 M854K is constitutively active in a cell-based IFN-β reporter assay. a** IFN-β reporter signaling assay in HEK cells. Luciferase activity is normalized against WT MDA5 + Poly(I:C) as 100%. $n = 6$ distinct samples. Lower panel: Western blots showing the expression level of the human MDA5 mutants in HEK cells. The FLAG tag on each mutant was detected with an anti-FLAG antibody. "−", without Poly(I:C) induction; "+", with Poly(I:C) induction. **b** IFN-β reporter assay in ADAR1-KO HEK cells. Lower panel: Western blots showing the expression level of the human MDA5 mutants in ADAR1-KO cells. $n = 3$ distinct samples. **c** IFN-β reporter assay with titration of protein expression level in HEK cells. ¼, 1, and 4 indicate the fold of plasmids transfected. ns, $P > 0.05$; **,$P < 0.01$; ***$P < 10^{-3}$ ($n = 3$ distinct samples, ordinary one-way ANOVA). WT MDA5 shows a switch-like "On-Off" activation mode for signaling. The mutants show signal activation proportional to their expression level, at low dsRNA abundance without poly(I:C). Lower panel: Western blots showing the expression level of the human MDA5 mutants in titration experiments. All error bars in this figure represent SEM between measurements, centered on the mean. Source data are provided as a Source Data file. All cell-based assays were repeated independently at least three times with similar results.

from conformational changes coupled to ATP hydrolysis, which promote dissociation from RNAs with imperfect duplexes[14]. A-to-I deamination by ADAR1 weakens base pairing in Alu:Alu mRNA duplexes to prevent autoinflammatory recognition by MDA5[19]. The signaling activity of the M854K variant in unstimulated cells was approximately the same as that of WT MDA5 in cells stimulated with poly(I:C) (Fig. 1a). This suggests that, like previously studied disease-associated MDA5 variants, the M854K variant forms filaments on the relatively short duplexes within cellular RNA[19]. To confirm the importance of the filament interface in activating MDA5 signaling from cellular RNA, we measured the signaling activities of our panel of MDA5 variants in ADAR1-knockout (ADAR1-KO) cells. The mutant with a stabilized filament interface, H871A/E875A, was constitutively active in ADAR1-KO cells, with poly(I:C) stimulation providing no further increase in signaling (Fig. 1b). The mutant fully deficient in filament formation, D848K/F849A/R850E,

remained fully deficient in signaling, but the partially deficient mutant, ΔC12, gained some basal signaling activity (25% of WT) without poly(I:C) stimulation in ADAR1-KO cells (Fig. 1b). Together, these data indicate that M854K and other gain-of-function disease mutations promote MDA5 helical filament assembly on endogenous cytosolic RNAs to the extent that the stability or half-life of the filaments becomes sufficient to activate interferon signaling.

The constitutive signaling activity of variant M854K in the absence of exogenous long dsRNAs implies that the M854K mutation disrupts the ATPase-dependent structural selectivity of MDA5 for long uninterrupted RNA duplexes. Cellular dsRNAs are relatively short. Alu:Alu RNA duplexes span up to 300 bp but contain a mismatch or bulge every 13 bp on average. We hypothesized MDA5 signaling from cellular RNAs to be less sensitive and cooperative than signaling from long viral dsRNA. To test this, we measured the cell signaling activity of the WT,

M854K, and ΔC12 variants with different expression levels of each variant. As expected, WT MDA5 had no significant signaling activity without poly(I:C) stimulation, regardless of MDA5 expression level, and high signaling with poly(I:C) stimulation. Signaling of WT with poly(I:C) increased slightly, but not proportionally with increasing MDA5 expression levels (Fig. 1c). With the M854K variant, however, higher protein expression levels produced higher signaling activity, particularly without poly(I:C) stimulation, when dsRNA was less abundant. The loss-of-function interface mutant ΔC12 had no signaling activity without stimulation and proportionally higher signaling with higher protein levels with poly(I:C) induction. Doubling the amount of luciferase vector-transfected had no significant effect on measured signaling activity, regardless of MDA5 expression level (Supplementary Fig. 1b), indicating that the amount of luciferase-reporter vector was not limiting. In conclusion, WT MDA5 has an ultrasensitive and cooperative signaling ("On-Off") response that is selective for long dsRNA. M854K and ΔC12 variants produce more proportional, less cooperative responses, with M854K lacking selectivity and ΔC12 lacking sensitivity.

We also measured signaling activity in cells stimulated with linear 100-bp, 300-bp, and 1-kbp in vitro-transcribed dsRNA. The signaling activity of M854K MDA5 with each of these dsRNAs or with no stimuli was similar as when transfected with poly(I:C) (Supplementary Fig. 1c). In contrast, signaling of WT was significantly smaller with all three linear dsRNAs than with poly(I:C). Overall, this supports the conclusion that M854K MDA5 is less selective than WT MDA5, and that in the cellular context the M854K variant is activated by endogenous RNA even without stimulation with exogenous RNA ligands.

**MDA5 variant M854K lacks ATPase activity and has increased affinity for dsRNA.** The ATPase activity of MDA5 promotes dissociation from the short (<300-bp) dsRNAs present in the cytosol such as Alu:Alu hybrids. Our cell signaling data imply that, like other gain-of-function disease-associated MDA5 mutations[26], the M854K mutation stabilizes MDA5-RNA complexes with endogenous RNAs to an extent that the complexes activate signaling. Biochemical analyses of gain-of-function variants have shown that mutations in the ATP binding site such as R337G increase the affinity of MDA5 for short dsRNAs in the presence of ATP by inhibiting ATP hydrolysis and hence ATP-dependent proofreading[25,26]. Mutations mapping to the RNA binding interface, such as D393V and G495R, increase the intrinsic affinity of MDA5 for dsRNAs[19,26,30]. To understand the biochemical consequences of the M854K mutation, we purified the mutant protein (Supplementary Fig. 1d, e) and measured its ATPase, dsRNA binding, and filament-forming activities. We note that purified M854K MDA5 precipitated after storage on ice for 4–7 days, whereas WT MDA5 remained soluble, suggesting the M854K variant was less stable. We found that the M854K variant had no detectable ATPase activity (Fig. 2a). To assess RNA binding affinity, we incubated increasing concentrations of MDA5 with RNA, added ATP, and analyzed complex assembly in native electrophoretic mobility shift assays (EMSAs). The dsRNA used for this assay was a 310-nt (ca. 150-bp) Alu(+):Alu(−) dsRNA hybrid from the NICN1 3′UTR known to support IRF3 dimerization in vitro in the context of ADAR1 deficiency or MDA5 gain-of-function mutations[19]. At the lowest M854K MDA5 protein concentration tested of 200 nM, we found that the dsRNA migrated primarily as a saturated complex with M854K MDA5 whereas with WT MDA5 the dsRNA migrated as a smeared band (Fig. 2b). This indicates that M854K MDA5 binds Alu(+):Alu(-) dsRNA with a slightly higher affinity than WT MDA5 (Fig. 2b). At higher protein concentrations, a fraction of

WT MDA5-dsRNA complexes failed to enter the gel or migrated as a high-molecular-weight smear, suggesting the complexes were crosslinked or aggregated. We note that M854K and WT MDA5 bound Alu(+) ssRNA with approximately equal affinity, only slightly lower than the affinity of WT MDA5 for the dsRNA hybrid (Fig. 2c). The binding of MDA5 to ssRNA has been reported[4], but there is no evidence ssRNA binding activates signaling.

Negative-stain electron microscopy showed M854K MDA5 formed filaments on Alu(+):Alu(−) dsRNA and 100-bp dsRNA in the presence of ATP and failed to form filaments without dsRNA (Fig. 2d). WT MDA5 formed fewer, shorter filaments, along with some aggregates, in the presence of Alu(+):Alu(−) dsRNA (two versus 22 filaments observed in six micrographs of WT and M854K MDA5, respectively) and none with 100-bp dsRNA or in the absence of dsRNA (Fig. 2d). Together, these results support the model that the lack of ATPase activity of the M854K variant allows it to form filaments on Alu(+):Alu(−) and 100-bp dsRNA by reducing the rate of filament disassembly, which is driven by ATP hydrolysis. The increased stability of the filaments explains the increased signaling activity of the M854K mutant.

**Lys854 forms polar bonds with residues in Hel1 and Hel2 in ATP-bound MDA5 filaments.** Unlike previously described disease mutations, which map to the ATP and RNA binding sites, the M854K mutation is in the pincer domain. To gain a mechanistic understanding of the effects of the M854K mutation on MDA5 function, we determined structures of mouse M854K MDA5-dsRNA filaments at different stages of ATP hydrolysis by cryoEM image reconstruction. Helical reconstructions of filaments with RELION3.1[31] produced maps sufficiently detailed for atomic models to be built and refined (Table 1). The highest-resolution structure, with an overall resolution of 2.8 Å and local resolutions up to 2.7 Å, was obtained from ATP-bound filaments (Fig. 3a–c and Supplementary Fig. 3). This structure provided more accurate and precise views of the side chains, ATP, RNA, and solvent molecules than previously reported MDA5 structures (Supplementary Fig. 4). The ATP-bound M854K MDA5-dsRNA filaments have essentially the same overall structure and helical symmetry as WT MDA5-dsRNA filaments with ATP bound. The Hel1 and Hel2 domains are in the semi-closed state (as defined previously[12]) and the helical twist of 75° is low (as defined previously[14]). With a helical rise of 44 Å, the asymmetric unit spans 14 bp of dsRNA with a protein–RNA interface area of 2168 Å$^2$ similar to WT ATP-bound filaments. The M854K variant binds ATP in the same manner as WT MDA5, with a magnesium ion coordinating the β- and γ-phosphate groups. The most notable difference in the ATP-bound M854K structure is that the mutant Lys854 side chain forms a salt bridge with Glu813 in Hel2, and a hydrogen bond with Ser491 in Hel1, within the same MDA5 protomer (Fig. 3d–f). These polar bonds are more constrained in distance and orientation than the weak hydrophobic contacts formed by the Met854 sidechain in WT MDA5 (Supplementary Fig. 5a, b). Thus, the M854K mutation imposes constraints absent in WT, between the pincer and Hel1/Hel2 domains.

To assess the contributions of the polar contacts between Lys854, Ser491, and Glu813 to inhibition of ATP hydrolysis, we measured the ATPase activities of M854K MDA5 proteins in which Ser491 or Glu813 were mutated to alanine to preclude polar sidechain contacts with Lys854. The S491A/M854K and E813A/M854K mutants remained inactive, but the triple mutant S491A/E813A/M854K recovered ATPase activity to a level comparable to that of WT MDA5 (Fig. 3g). Hence, the polar

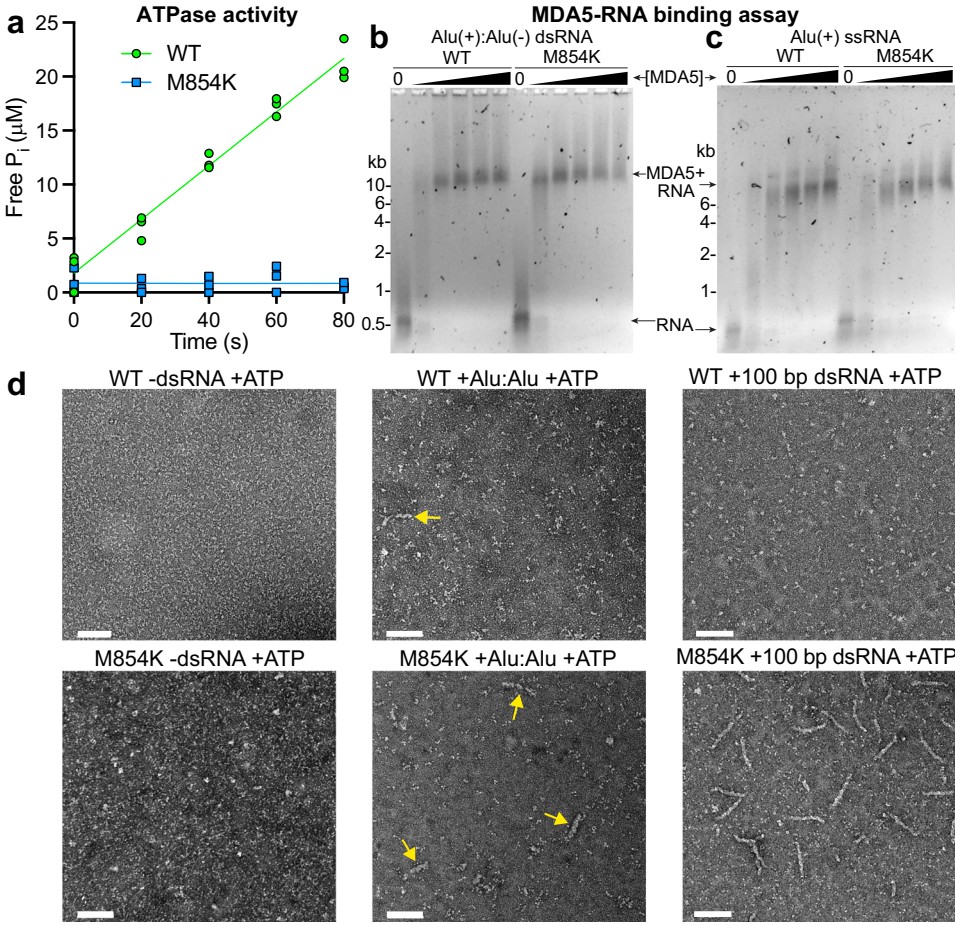

**Fig. 2 The ATPase, RNA binding, and filament-forming activities of MDA5 M854K. a** ATPase assay of WT MDA5 and M854K MDA5 with 1 kb dsRNA. The assay was performed in triplicate with three independently purified batches of MDA5 protein. **b** Gel-shift assay (EMSA) of WT or M854K MDA5 (0, 0.2, 0.4, 0.6, 0.8, 1 μM) mixed with 5 ng/μl annealed Alu(+):Alu(−) dsRNA, followed by addition of 6 mM ATP. Agarose gels were stained for RNA with SYBR Gold. The M854K variant binds to Alu(+):Alu(−) with higher affinity than WT MDA5 with ATP present. **c** EMSA of increasing concentrations of WT or M854K MDA5 after incubation with 5 ng/μl Alu(+) ssRNA performed under the same conditions as in (**b**). The M854K variant binds ssRNA with a similar affinity as WT MDA5 with ATP present. **d** Negative-stain electron micrographs of M854K and WT MDA5 with ATP and Alu(+):Alu(−) dsRNA, 100-bp dsRNA, or without dsRNA. Arrows indicate MDA5-Alu:Alu complexes. Scale bar, 100 nm. Micrographs are representative of at least six images collected for each condition (see Supplementary Fig. 2 for additional micrographs). ATPase assays and EMSAs were repeated independently at least three times with similar results. Source data are provided as a Source Data file.

contacts with the Ser491 and Glu813 side chains both contribute to inhibiting ATP hydrolysis by M854K MDA5, and contact with either of these side chains is sufficient to prevent ATP turnover.

**M854K partially uncouples transition to the closed state from RNA footprint expansion.** In the first step of MDA5 catalysis, the Hel1 and Hel2 domains move from the semi-closed ATP-bound catalytic ground state to the closed transition state, bringing the nucleotide-binding motifs in Hel2, motifs Va and VI, into position for catalysis. Only in this closed state are all six nucleotide-binding helicase motifs engaged with the nucleotide[9,12,14]. The transition to the closed state is transduced by the pincer domain and coupled to an increase in the RNA binding footprint from 14 to 15 bp per MDA5 molecule[14]. CryoEM image reconstructions of M854K MDA5-dsRNA filaments bound to transition state analog ADP-AlF₄ produced two different structures of equal overall resolution at 4.3 Å (Fig. 4, Supplementary Fig. 6 and Table 1). Both structures had clear density for ADP-AlF₄ (distinguishable from ADP), closed helicase motifs, and intermediate helical twists (87–88°), like WT-ADP-AlF₄-bound filaments[14]. However, the commonly detected structure, Class 167k, with 166,554 segments collected, contained

only 14 bp of RNA per MDA5 molecule, the same number as ATP-bound filaments but one base pair less than WT-ADP-AlF₄-bound filaments. In the less common structure, Class 62k (62,028 segments), the footprint was 15 bp (Fig. 4b). Hence, in a majority of MDA5 protomers, the M854K mutation inhibits the RNA footprint expansion normally coupled with progression from the semi-closed ground state to the closed transition state. RNA footprint expansion is a key feature of the conformational change that accompanies ATP hydrolysis, which has been proposed to perform a proofreading function in discrimination of self- versus non-self dsRNA[14]. Local footprint expansion within MDA5-RNA filaments during ATP hydrolysis has also been proposed to allow the repair of MDA5 filament discontinuities and displacement of viral proteins from dsRNA[14,32,33]. The M854K mutation interferes with each of these functions by partially uncoupling the transition to the closed state from RNA footprint expansion, possibly due to the constraints imposed by the M854K mutation across the He1/Hel2/pincer domain interface.

We note that the dominant ADP-AlF₄-bound filament reconstruction incorporated three times more segments than the equivalent WT reconstruction[14], but this M854K reconstruction

**Table 1 CryoEM data collection, structure determination, and model refinement parameters.**

| | M854K MDA5-dsRNA 10 mM ATP | M854K MDA5-dsRNA 1 mM ADP-AlF$_4$ | | WT MDA5-dsRNA 2 mM ADP | |
|---|---|---|---|---|---|
| **Data collection and processing** | | | | | |
| Microscope | 300 kV Titan Krios | 300 kV Titan Krios | | 300 kV Titan Krios | |
| Electron exposure (electrons Å$^{-2}$) | 40.0 | 29.6 | | 44.5 | |
| Exposure per frame (electrons Å$^{-2}$) | 1.00 | 0.49 | | 1.11 | |
| Nominal defocus range (μm) | −1.2 to −2.5 | −0.5 to −3.5 | | −0.5 to −3.0 | |
| Pixel size (Å) | 0.86 | 1.084 | | 1.04 | |
| N. initial segment images | 1,068,108 | 731,263 | | 1,063,529 | |
| 3D class average | 854K-ATP | 854K-AlF$_4$-167k | 854K-AlF$_4$-62k | WT-ADP-88° | WT-ADP-92° |
| N. final segment images | 383,128 | 166,554 | 62,028 | 75,292 | 64,992 |
| Map resolution range (Å) | 190–2.8 | 240–4.26 | 240–4.26 | 241–3.89 | 241–3.35 |
| FSC threshold for resolution limit | 0.143 | 0.143 | 0.143 | 0.143 | 0.143 |
| Max. local resolution range | 4.8–2.69 | 8.5–3.8 | 9.1–3.9 | 8.5–3.55 | 5.8–3.2 |
| **Model refinement** | | | | | |
| Map sharpening B factor (Å$^2$) | −50 | −90 | −100 | −100 | −60 |
| Helical twist (°) | 74.63 | 87.3693 | 88.0297 | 88.0249 | 91.7469 |
| Helical rise (Å) | 43.92 | 44.4029 | 45.0795 | 44.2559 | 44.6204 |
| Mask correlation coefficient | 0.83 | 0.78 | 0.76 | 0.76 | 0.78 |
| **Model composition** | | | | | |
| N. non-hydrogen atoms | 6162 | 6012 | 6165 | 5890 | 5810 |
| Protein residues | 689 | 673 | 683 | 650 | 640 |
| RNA nucleotides | 28 | 28 | 30 | 30 | 30 |
| Ligand | ATP | ADP-AlF$_4$ | ADP-AlF$_4$ | ADP | ADP |
| Ions (Zn$^{2+}$, or Zn$^{2+}$, and Mg$^{2+}$) | 2 | 1 | 1 | 1 | 1 |
| **R.m.s. deviations** | | | | | |
| Bond lengths (Å) | 0.009 | 0.009 | 0.008 | 0.006 | 0.010 |
| Bond angles (°) | 0.758 | 0.862 | 0.734 | 0.651 | 0.853 |
| **B-factors/ADPs** | | | | | |
| Minimum | 21 | 40 | 81 | 94 | 71 |
| Maximum | 73 | 176 | 214 | 215 | 167 |
| Mean | 41 | 110 | 147 | 152 | 113 |
| **Validation** | | | | | |
| MolProbity overall score | 1.44 | 2.15 | 2.14 | 1.67 | 1.74 |
| MolProbity all-atom clashscore | 2.28 | 14.6 | 13.6 | 9.80 | 6.92 |
| Rotamer outliers (%) | 0.32 | 0.84 | 0.82 | 0.52 | 0.93 |
| **Ramachandran plot** | | | | | |
| % Favored | 93.2 | 92.3 | 91.7 | 97.2 | 94.9 |
| % Allowed | 6.5 | 7.2 | 8.1 | 2.8 | 5.0 |
| % Outliers | 0.3 | 0.5 | 0.2 | 0 | 0.1 |
| PDB code | 7BKP | 7NIQ | 7NIC | 7NGA | 7BKQ |
| EMDB code | EMD-12213 | EMD-12288 | EMD-12294 | EMD-12092 | EMD-11937 |
| EMPIAR code | 10630 | 10664 | 10664 | 10653 | 10653 |

*ADPs* atomic displacement parameters.
See also Figs. 3, 4, and 5 and Supplementary Figs. 3, 6, and 7.

had lower overall and local resolutions (Supplementary Fig. 6c, d) and slightly broader twist distribution (Fig. 4c), suggesting that M854K filaments are structurally more heterogeneous or flexible than WT filaments in the transition state. Additionally, Arg822 and Arg824, which coordinate the nucleotide β- and γ-phosphates in the WT MDA5-dsRNA transition state[14], do not appear to coordinate the nucleotide in the ADP-AlF$_4$-bound M854K MDA5 structures, although this could be attributable to the relatively low resolution of the maps.

**Hel2 motifs that bind ATP and contact Lys854 are disordered in ADP-bound WT MDA5.** The partial inhibition of RNA footprint expansion by the M854K mutation is expected to interfere with key functions of MDA5 as described above, but this does not fully explain the M854K variant's lack of ATPase activity. We have shown above that the variant can still bind ATP and reach the catalytic transition state, suggesting that the M854K mutation additionally inhibits subsequent steps in catalysis. Structures of MDA5-dsRNA filaments have been deter-

mined with ATP or ADP-AlF$_4$ bound and without nucleotide[14], but not in complex with ADP. To complete our structural picture of the MDA5 ATPase cycle, we determined cryoEM structures of ADP-bound WT MDA5-dsRNA filaments in the presence of magnesium (at a physiological concentration). We obtained two similarly populated structural classes, one with an intermediate helical twist (88°) and the other with a high twist (92°), with overall resolutions of 3.9 and 3.35 Å, respectively (Fig. 5a, Supplementary Fig. 7, and Table 1). In both structures, the density in the nucleotide-binding pocket unambiguously corresponds to ADP (or ADP:Mg$^{2+}$; Fig. 5b, c). The filament structure and helical symmetry are similar to those of nucleotide-free filaments, with the helicase in the semi-closed state, and a 2018 Å$^2$ protein-RNA interface area (smaller than the ground and transition states). However, we noticed that density in the map was weak for residues 810–827, to the extent that it was uninterpretable in the (higher-resolution) high-twist structure (Fig. 5b and Supplementary Fig. 5e). This indicates that residues 810–827 are more mobile or disordered in the ADP-bound state.

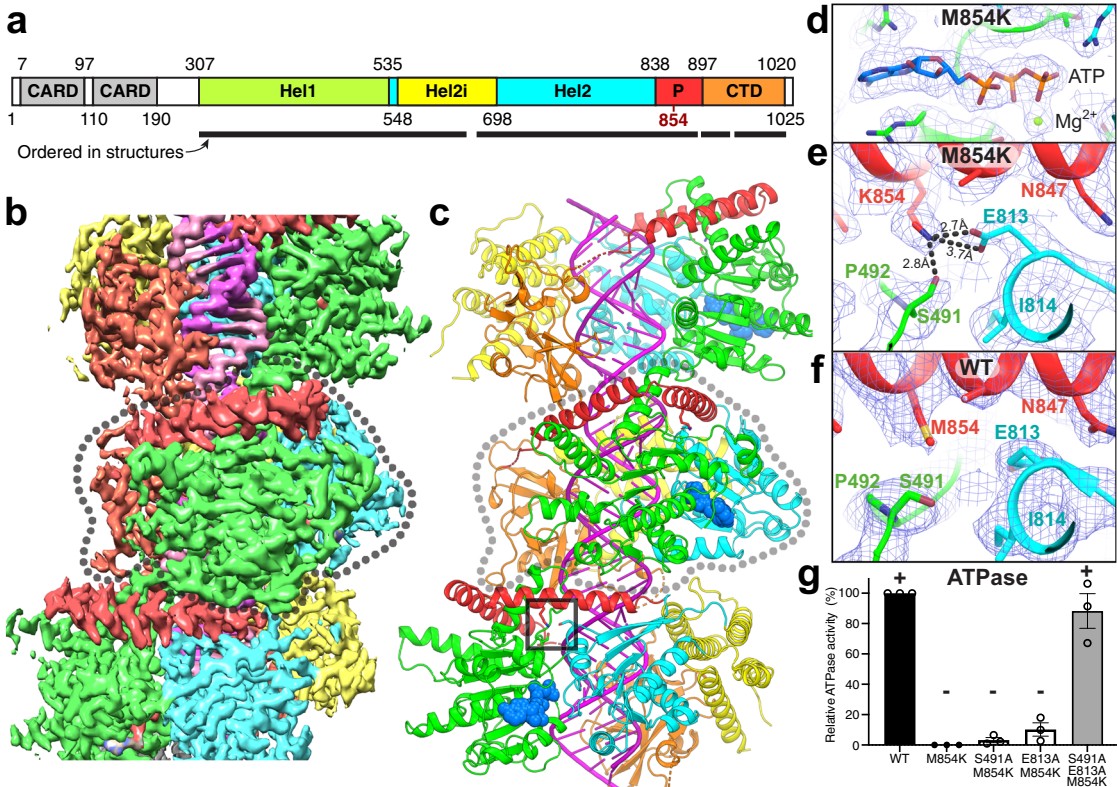

**Fig. 3 CryoEM structure of the ATP-bound M854K MDA5-dsRNA filament. a** Domain architecture of human/mouse MDA5. CARD caspase recruitment domain; Hel1 and Hel2 first and second RecA-like helicase domains; Hel2i, Hel2 insert domain; P pincer domain; CTD C-terminal domain. The same color code is used in panels (**b–e**), Fig. 5a and Supplementary Fig. 4. **b** 3D density map of the ATP-bound M854K MDA5-dsRNA filament at 2.8 Å overall resolution, colored by domain as in (**a**), with RNA in magenta. An MDA5 protomer is outlined in dashed gray. **c** Overview of the atomic model of the ATP-bound M854K MDA5-dsRNA filament. The central protomer of three MDA5 protomers is outlined in light gray. ATP is shown in sphere representation. The region surrounding residue 854 is boxed. **d**, **e** Closeups of cryoEM density maps: the bound ATP, **d**, and the region surrounding residue 854, showing polar contacts formed by Lys854 with residues from the Hel1 and Hel2 domains, (**e**). Polar contacts formed by Lys854 are shown as dashed lines. **f** Closeup of the region surrounding residue 854 in the previously reported structure of the WT MDA5-dsRNA filament with AMPPNP bound[14]. **g** Relative ATPase activities of S491A/M854K, E813A/M854K, and S491A/E813A/M854K mutants, normalized to WT (100%) and M854K (0%). Error bars represent SEM between measurements, centered on the mean. ($n = 3$ independent samples; $t = 15$ min). ATPase assays were repeated in independent experiments three times with similar results. Source data are provided as a Source Data file.

This is significant because this region spans conserved Hel2 helicase motifs Vc (residues 810–815) and VI (residues 817–826), which bind RNA and nucleotide, respectively[12], along with Glu813, the residue that forms a salt bridge with Lys854 in the ATP-bound structure of the M854K variant. Notably, motif VI contains Arg822 and Arg824, which coordinate the phosphate groups of the transition state nucleotide[14], and variant R822Q is associated with autoinflammatory disease[25,34]. Density for motifs Vc-VI, including Glu813, is similarly weak in the nucleotide-free state but ordered in ground and transitions states, including in filament structures determined at lower resolution[14] (Fig. 5b–e and Supplementary Fig. 5). This suggests that, in WT MDA5, motifs Vc-VI become more conformationally dynamic as catalysis proceeds from the transition state to the post-hydrolysis state, thereby contributing to MDA5 relaxing its grip on both the dsRNA and the nucleotide. Together with the decrease in total MDA5-RNA contact area that accompanies ATP hydrolysis[12,14], this could promote dissociation of ADP and imperfect RNA duplexes. In the M854K variant, however, the salt bridge between Lys854 and Glu813 (located in motif Vc), may constrain motifs Vc-VI and hence reduce dissociation of dsRNA and nucleotide, in association with the hydrogen bond between Lys854 and Ser491 (Fig. 3g).

**Mutation M854K alters the helical twist distribution of MDA5 filaments.** ATP hydrolysis by MDA5 requires a rotation of Hel2 relative to Hel1, which results in an increase in RNA footprint and filament twist[14]. Our structural data suggest that the M854K mutation imposes conformational constraints on the Hel1 and Hel2 domains, partially inhibiting RNA footprint expansion. To assess the effect of the M854K mutation on filament twist, we analyzed the helical twist distributions of filament segments by performing 3D classifications on each cryoEM dataset. We found that with ATP bound all M854K MDA5-dsRNA filament segments had a low helical twist (71°–81°). The twist distribution was narrower than for ATP-bound WT MDA5 filament segments, of which 30% had an intermediate twist (81°–91°) (Fig. 6 and Supplementary Fig. 8). The narrower twist distribution is consistent with the M854K mutation constraining the relative orientations of the Hel1, Hel2, and pincer domains.

Conversely, the twist distribution of ADP-AlF$_4$-bound filament segments was broader for M854K MDA5 than for WT. Whereas WT transition-state filament segments had an intermediate twist and an expanded 15-bp RNA footprint, 25% of M854K transition-state segments had high twist (91°–96°) (Fig. 6 and Supplementary Fig. 8). As noted above, a majority of M854K filament segments failed to expand their RNA binding footprint

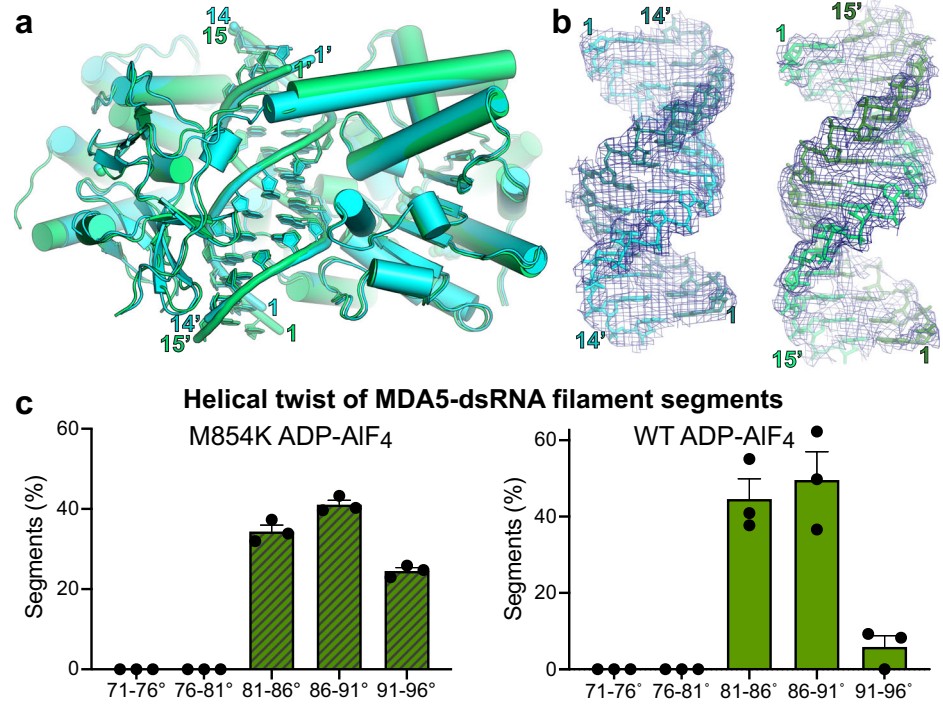

**Fig. 4 CryoEM structures of ADP-AlF$_4$-bound M854K MDA5-dsRNA filaments. a** Superposition of two atomic models of ADP-AlF$_4$-bound M854K MDA5-dsRNA filaments representing the most populated classes of 3D cryoEM image reconstructions. The class representing the majority of filament segments (Class 167k, cyan) contains 14 bp RNA in the asymmetric unit; the minority structural class (Class 62k, green) contains 15 bp RNA. **b** Atomic models and 3D density maps for the dsRNA for the two structural classes of ADP-AlF$_4$-bound M854K MDA5-dsRNA filaments. Left, Class 167k, 6.8-σ contour level in PyMol; Right, Class 62k, 6.0-σ contour level. **c** Histograms showing the distributions of filament segments as a function of helical twist for M854K and WT MDA5-dsRNA with ADP-AlF$_4$ bound. The twist distribution for WT is shown for reference and was reported previously[14]. The distributions are from 3D classifications performed with ten classes per dataset. Error bars represent the SEM between 3D classification calculations, centered on the mean ($n = 3$ independent calculations). Source data are provided as a Source Data file.

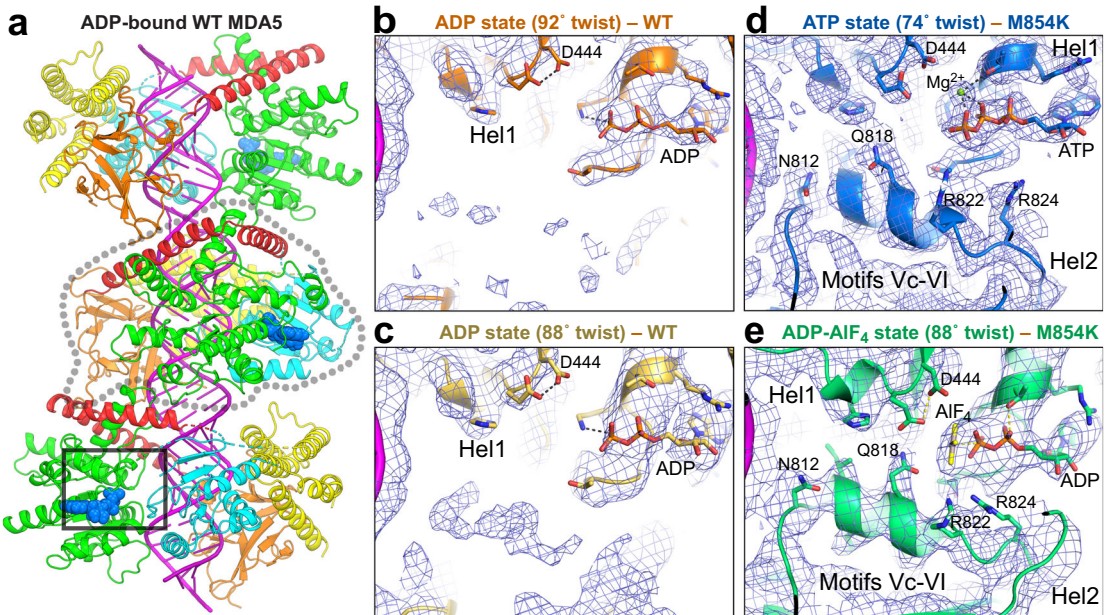

**Fig. 5 CryoEM structures of WT MDA5-dsRNA filaments in complex with ADP. a** Atomic model of the WT MDA5-dsRNA filament with ADP bound and 92° helical twist, determined from a cryoEM map at 3.35 Å overall resolution. The central protomer of three MDA5 protomers is outlined in dashed gray. The region around the ATP binding site and motifs Vc-VI, shown in (**b**), is boxed. The color code is the same as in Fig. 1. **b, c** CryoEM density at the Hel1-Hel2 interface in ADP-bound WT MDA5-dsRNA filaments. **b** High-helical twist (92°) structure; **c** intermediate-twist (88°) structure. **d, e** CryoEM density at the Hel1-Hel2 interface in ATP-bound M854K MDA5-dsRNA filaments with an Mg$^{2+}$ ion shown as a green sphere, **d**, and ADP-AlF$_4$-bound M854K MDA5-dsRNA filaments (Class 62k), **e** Motifs Vc-VI become disordered in the transition from the ADP-AlF$_4$ state to the ADP state. The same contour level of 5.5 σ in PyMol was used for panels **b**–**e**.

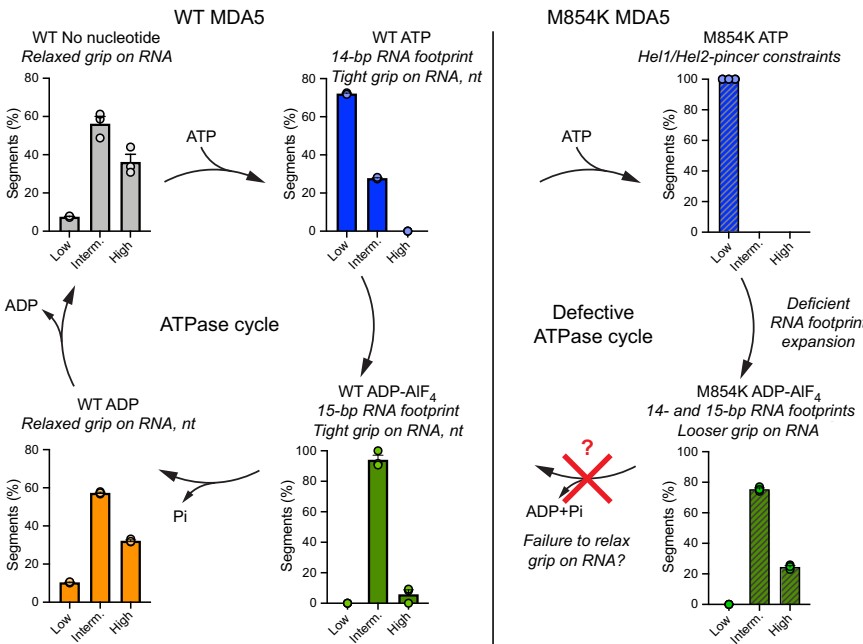

**Fig. 6 Helical twist of MDA5 filaments in the ATPase cycle and proposed effect of M854K on the twist.** Histograms showing the distributions of 3D cryoEM reconstructions as a function of helical twist for WT and M854K MDA5-dsRNA filaments with ATP, ADP-AlF$_4$, ADP, or no nucleotide bound. The distributions shown are from 3D classifications performed with ten classes per dataset. Error bars represent SEM between 3D classification calculations, centered on the mean ($n = 3$ independent calculations). The histograms for nucleotide-free, ATP-bound, and ADP-AlF$_4$-bound WT MDA5-RNA filaments are from data reported previously[14]. Low, twist = 71°–81°; Interm., twist = 81°–91°; High, twist = 91°–96°. "nt", ANP nucleotide. See also Supplementary Fig. 8. Source data are provided as a Source Data file.

and retained the 14-bp footprint of the ATP-bound ground state. We conclude that the M854K mutation partially inhibits RNA footprint expansion of MDA5 as the helicase domains transition to their closed transition state.

In the ADP-bound state, WT MDA5-RNA filament segments had the same broad distribution of low, intermediate, and high twists as in the nucleotide-free state, with intermediate twists being the most abundant (Fig. 6 and Supplementary Fig. 8). These broad twist distributions, along with the reduced protein-RNA contact area relative to the ground state, are suggestive of more flexible or relaxed states, in which MDA5 has loosened its grip on bound RNA and nucleotide.

## Discussion

This work reveals the structural basis and functional deficiencies of the M854K MDA5 variant associated with severe autoinflammatory disorders. Our RNA binding and cell signaling assays imply that the M854K mutation increases the stability of MDA5 filament assemblies on cytosolic RNAs, including inverted Alu repeats, to the extent that interferon signaling is constitutively activated. This explains the elevated interferon signaling in patients harboring the mutation without viral infection[28]. Additionally, the M854K variant produces a much less cooperative signaling response than WT MDA5, resulting in signaling proportional to the amount of MDA5 and RNA present rather than an ultrasensitive "On-Off" response. Despite its location in the pincer domain, outside the RNA interface and ATP binding site, the M854K mutation abolishes ATPase activity. Since ATP hydrolysis promotes dissociation of MDA5 from imperfect RNA duplexes, we propose that the M854K variant induces autoinflammatory signaling due to a lower dissociation rate from endogenous RNAs. The net effect of M854K on signaling is similar to that of disease mutations in the ATP or RNA binding sites[26], or loss of ADAR1 activity[19].

Our cryoEM structure of the M854K-dsRNA filament with ATP bound contains structural information up to 2.7 Å resolution, significantly higher than previously published MDA5 filament structures. The structure shows that Lys854 forms polar contacts with the Hel1 and Hel2 domains, imposing constraints on the distance and relative orientation of these two domains. Consistent with this, the filaments had a narrower twist distribution than WT MDA5 filaments with ATP bound. In cryoEM reconstructions of the ADP-AlF$_4$ transition state, most of the M854K MDA5 filament protomers fail to expand their RNA footprint from 14 to 15 base pairs. The twist distribution of ADP-AlF$_4$-bound filaments was also broader for M854K MDA5 than for WT. Together, these findings suggest that the M854K mutation increases structural heterogeneity in the transition state. RNA footprint expansion has been proposed to contribute to discrimination between self- and non-self RNAs, repair of filament discontinuities, and displacement of viral proteins from dsRNA[14,32,33]. By partially inhibiting RNA footprint expansion, the M854K mutation will interfere with each of these functions.

The structure of ADP-bound WT MDA5 filaments completes our structural picture of the MDA5 ATPase cycle (Fig. 6). In contrast to the ATP- and transition states, helicase motifs Vc and VI in Hel2 are mostly disordered in the ADP state. Motif Vc contains Glu813, the residue that forms a salt bridge with Lys854 in the ATP state of the M854K variant. Motif VI contains Arg822 and Arg824, which directly coordinate the nucleotide in the transition state. Notably, variant R822Q is associated with elevated interferon signaling and with Singleton–Merten Syndrome[25,34]. Motifs Vc and VI becoming more conformationally dynamic as catalysis proceeds from the transition state to the post-hydrolysis state could cause MDA5 to relax its grip on the dsRNA and ADP. Together with the decrease in total MDA5-RNA contact area that accompanies ATP hydrolysis[12,14], this could promote dissociation of imperfect (endogenous) RNA duplexes. In the M854K variant, however, the salt bridge between

Lys854 and Glu813, located in motif Vc, may constrain motifs Vc-VI, and hence prevent dissociation of certain endogenous dsRNAs and of ADP/Pi.

The structural data in this study provide insights into the role of ATP hydrolysis in RNA recognition by MDA5. In the ATP and transition states, MDA5 has a tight grip on the RNA, as evidenced by the larger protein-RNA interface, narrow twist distribution, and ordered motif Vc-VI structure in these two states. Upon ATP hydrolysis, a reduction in the RNA binding surface, broadening of the twist distribution, and disordering of motif Vc-VI cause MDA5 to relax its grip on the RNA. This in turn provides an opportunity for dissociation of RNAs with lower binding affinities, such as endogenous RNAs. In the M854K variant, polar contacts between the Lys854 sidechain and the Hel1 and Hel2 domains constrain some of the conformational changes necessary for ATP hydrolysis, including those required for RNA footprint expansion. This interferes with ATP-dependent RNA discrimination or proofreading and ultimately leads to constitutive signaling from endogenous RNAs.

Our study uncovers the structural basis and functional consequences of the M854K disease-associated MDA5 variant. The M854K mutation inhibits ATP hydrolysis through an allosteric mechanism, setting this variant apart from previously studied ATPase-deficient variants, which directly target nucleotide binding or catalysis in the active site pocket. By completing our picture of the MDA5 ATPase cycle, our structural data also provide insights on the structural changes driven by ATP hydrolysis that allow MDA5 to distinguish between self and non-self RNA. A clearer understanding of how genetic variants dysregulate these processes will help identify new immunotherapeutic strategies.

## Methods

**MDA5 protein purification**. Genes encoding mouse *MDA5 (IFIH1)* were cloned into the pET28a vector with an N-terminal hexahistidine tag followed by a tobacco etch virus (TEV) protease cleavage site as described[5]. MDA5 residues 646–663, in the flexible L2 surface loop of Hel2i, were deleted for solubility, resulting in a 114-kDa polypeptide chain. The ΔL2 loop deletion does not affect the dsRNA binding, ATPase, or interferon signaling activities of MDA5[5,13,14]. The M854K and other mutations were introduced into the MDA5-ΔL2 constructs by overlap PCR.

*Escherichia coli* BL21(DE3) cells were transformed with an *MDA5* construct and grown to $OD_{600}$ 0.4–0.6 at 37 °C. After cold shock on ice water for 30 min, protein expression was induced with 0.4–0.5 mM isopropyl-b-D-1-thiogalactopyranoside (IPTG) overnight at 22 °C. Harvested cells were resuspended in 50 mM HEPES 7.5, 0.15 M NaCl, 5% Glycerol, 20 mM imidazole, EDTA-free Complete Protease Inhibitor (Roche), and 8 mM 2-mercaptoethanol (β-ME), then lysed by ultrasonication on ice. The lysate was spun at 15,000×g for 1 h. The supernatant was loaded onto Ni-NTA agarose (QIAGEN) and MDA5 was eluted with 50 mM HEPES 7.5, 0.15 M NaCl, 5% Glycerol, 0.3 M imidazole, and 8 mM β-ME. MDA5 in the eluate was further purified on a Resource Q anion exchange column (Cytiva) (buffer A: 20 mM HEPES 7.5, 0.1 M NaCl, 2 mM dithiothreitol (DTT); buffer B: 20 mM HEPES 7.5, 0.5 M NaCl, 2 mM DTT), and a Superdex 200 Increase 10/300 GL size-exclusion column (Cytiva) in 20 mM HEPES 7.5, 0.1 M KCl, 5 mM $MgCl_2$, and 2 mM DTT. For the ATP-bound M854K MDA5 dataset, the protein was expressed as above except for the following differences. The protein was expressed in *E. coli* Rosetta2(DE3) cells overnight at 17 °C. Cells were lysed in 30 mM HEPES pH 7.7, 0.15 M NaCl, 5% glycerol, 1 mM Tris(2-carboxyethyl)phosphine (TCEP), EDTA-free Complete Protease Inhibitor (Roche), and benzonase (Merck). The size-exclusion chromatography buffer was 30 mM HEPES pH 7.8, 0.15 M KCl, 1 mM DTT. Purified proteins were flash-frozen and stored at −80°, except for M854K MDA5 EM grid preparation, for which fresh size-exclusion column elution fractions were used.

**Luciferase-reporter cell signaling assay**. The ADAR1-knockout (KO) HEK293T cell line was a kind gift from Charles Rice (The Rockefeller University). WT or ADAR1-KO HEK293T cells in 12-well plates were transfected with 400 ng ml⁻¹ of firefly luciferase under the control of the IFN-β promoter (pIFN-Luc, Promega), 40 ng ml⁻¹ of Renilla luciferase under a constitutive promoter (pRL-TK, Promega), and 40 ng ml⁻¹ of pLEXm vector[35] containing either no insert, WT human MDA5-ΔL2 (pLEXm MDA5-Δ644-663), or mutant human MDA5-ΔL2. Transfections were performed with polyethylenimine (Sigma-Aldrich). After expression for 6 h, cells were transfected with poly(I:C) (Tocris Bioscience). After 24 h, cell lysates were prepared, and luciferase activity was measured using Promega assay

kits according to the manufacturer's instructions. Firefly luciferase activity was normalized against the co-transfected Renilla luciferase[14]. For cell signaling assays with short dsRNAs, 96-well plates containing $4 \times 10^4$ HEK293T cells were transfected with 80 ng pIFN-Luc, 8 ng pRL-TK, and 8 ng of pLEXm vector containing either no insert, human WT MDA5-ΔL2, or human M854K MDA5-ΔL2 using Lipofectamine™ 3000 Transfection Reagent (ThermoFisher). After 6 h incubation at 37 °C and 5% $CO_2$, cells were transfected with 100 ng of 100, 300 bp, or 1 kb in vitro-transcribed dsRNAs or with poly(I:C) using Lipofectamine™ MessengerMAX™ Transfection Reagent (ThermoFisher). For Western blots, the FLAG tag on MDA5 was detected with an anti-FLAG antibody (Sigma-Aldrich, cat. no. F1804, RRI-D:AB_262044,1:5000 dilution) or an anti-MDA5 antibody (Enzo Life Sciences, cat. no. ENZ-ABS299, RRID:AB_2893162, 1:1000 dilution). For the actin control, we used anti-actin antibody AC-40 (Abcam, cat. no. ab11003, RRID:AB_297660, 1:1000 or 1:2000 dilution).

**In vitro transcription of double-stranded RNA**. RNAs were transcribed in vitro using T7 RNA Polymerase (New England Biolabs) following the manufacturer's instructions. The Alu(+):Alu(−) hybrid used was the inverted repeat from Alu retrotransposon in 3′-UTR of the *NICN1* gene[19]. The Alu(+) and Alu(−) sequences were synthesized as separate gene fragments, which were used as templates to transcribe Alu(+) and Alu(−) in separate reactions. The 1-kb dsRNA contained the first 1 kb of the *IFIH1* gene flanked by 5′-GGGAGA and TCTCCC-3′. All transcripts were treated with DNase and purified by column-based purification (ThermoFisher PureLink RNA Mini Kit). To produce dsRNA or Alu(+):Alu(−) hybrids, transcripts were heated to 95 °C for 5 min then annealed by cooling to room temperature over at least 1 h.

**Electrophoretic mobility shift assays**. A total of 5 ng µl⁻¹ RNA was incubated on ice with purified MDA5 protein at the concentrations indicated in Fig. 2c, d in gel filtration buffer (with 5 mM $MgCl_2$) for 30 min. ATP was added to a final concentration of 6 mM and the solutions were incubated at 20 °C for 20 min. The reactions were quenched with native loading dye containing EDTA (New England Biolabs, Cat. no. #B7025S). Samples were loaded on a 0.7% agarose gel and run at 120 V in 45 mM (0.5×) Tris-borate buffer on ice for 40–60 min. The gels were post-stained for RNA with 1× SYBR Gold stain (ThermoFisher). Gels were imaged with a G-BOX gel imager (Syngene).

**ATPase assay**. Purified MDA5 protein was diluted to a concentration of 75 nM in a solution containing 2.25 nM 1 kb dsRNA, 1 mM ATP, 20 mM HEPES pH 7.8, 0.15 M KCl, 1.5 mM $MgCl_2$, 1 mM DTT, and incubated at 37 °C. Samples of the reaction were extracted and quenched with 20 mM EDTA at 20-s intervals. The concentration of inorganic phosphate released by hydrolysis of ATP was measured by tracking absorbance at 620 nm of malachite green binding to phosphate ions using the ATPase/GTPase activity colorimetric assay (Sigma-Aldrich) as described in ref. [14]. For the ATPase rescue assays (Fig. 3g), 140 nM of protein were incubated with 3.6 nM of 1 kb dsRNA for 15 min.

**Negative-stain EM**. For the filament assembly assay, 26 µg ml⁻¹ MDA5 protein was incubated with 3.24 µg ml⁻¹ of RNA in 20 mM HEPES pH 7.5, 0.1 M KCl, 5 mM $MgCl_2$, 6 mM ATP at room temperature for 30 min. Samples were applied to glow-discharged carbon-coated grids, negatively stained with uranyl acetate [2% (wt/vol)], and imaged with a 120 kV Technai 12 electron microscope (Thermo-Fisher). Images were taken at −2 to −4 µm defocus, 26,000× magnification, and with 4 Å per pixel.

**CryoEM sample preparation and data collection**. A total of 1 g l⁻¹ purified MDA5 protein was incubated with 0.05 g l⁻¹ 1-kb dsRNA in Buffer F (20 mM HEPES pH 7.7, 0.1 M KCl, 5 mM $MgCl_2$, 2 mM DTT) on ice for 2–3 min. For an optimal trade-off between filament quality and nucleotide-binding, 2 mM ADP or 1 mM ADP-AlF₄ (corresponding to a 200-fold or 100-fold molar excess over MDA5, respectively) was incubated with preformed MDA5-dsRNA filaments for 6–7 min (ADP dataset) or 30 min (ADP-AlF₄ dataset). Samples were diluted twofold with Buffer F and 3.5 µl of the sample was immediately applied onto a glow-discharged 300-mesh gold Quantifoil R1.2/1.3 grid (Quantifoil Micro Tools). Grids were blotted for 2–4 s and plunge-frozen in liquid ethane cooled by liquid nitrogen with a Vitrobot MarkIV (ThermoFisher) at 4 °C and 100% humidity. For ATP-bound MDA5 filaments, the sample was prepared as above except the protein was incubated with the dsRNA and 10 mM ATP in Buffer F2 (30 mM HEPES pH 7.8, 150 mM KCl, 1 mM DTT) on ice for 2–3 min. Samples were diluted twofold with Buffer F2 and applied to 300-mesh gold UltrAuFoil R1.2/1.3 grids, freshly glow-discharged for 60 s at 30 mA with an Edwards 12E6/531 glow discharger.

CryoEM imaging of ADP-bound WT MDA5-dsRNA filaments was performed at the CryoEM Service Platform at EMBL Heidelberg. Data were collected on a 300 kV Titan Krios microscope (ThermoFisher) with a K2 detector in counting mode. Movies were recorded at a magnification of 130,000 (1.04 Å pixel⁻¹) with a 12 s exposure time. Data were acquired with SerialEM with four shots per hole. ADP-AlF₄-bound M854K MDA5 filaments were imaged at the Electron Bio-Imaging Centre (eBIC, Harwell, UK) on a Titan Krios microscope with a Falcon3 detector in counting mode (ThermoFisher). Movies were recorded at a

magnification of 75,000 with a 60 s exposure time. Data were acquired with EPU (ThermoFisher) with two shots per hole. ATP-bound M854K MDA5 filaments were imaged at the MRC-LMB on a Titan Krios microscope with a K3 detector in counting mode (ThermoFisher). Movies were recorded with a 1.4 s exposure time. Data were acquired with EPU with two shots per hole. See Table 1 for additional data collection parameters.

**Image processing and helical reconstruction**. Movies were motion-corrected and dose-weighted with MotionCor2 in RELION3.1[31]. The contrast transfer function was estimated with CtfFind4[36] with non-dose-weighted and aligned micrographs. Image reconstruction was performed in RELION3.1[37].

For the ADP-bound WT MDA5-dsRNA structure, 1,063,529 segments were picked from 4680 movies. The extracted classes underwent nine rounds of 2D classification to remove poor segments. About 860,175 segments were used in 3D refinement, resulting in a map with twist 89.1° and rise 44.8 Å as a representative map for the whole dataset. The refined segments were further subjected to 3D classification. A 3D class with 72,133 particles at twist 90.3° and rise 44.6 Å was selected. The 3D class was subjected to further 3D classification and two 3D classes were selected, one with twist 91.9°, rise 44.8 Å and the other with twist 91.7° rise 44.8 Å, with a total of 64,992 segments. We further split the optics groups for the segments and performed particle polishing and CTF refinement[38,39]. Final 3D refinement yielded a 3.6 Å reconstruction with twist 91.7° and rise of 44.6 Å. Helical symmetry was imposed on the unfiltered half maps, and a B-factor of $-60$ Å$^{-2}$ was applied to the reconstruction. Postprocessing yielded the final map at 3.35 Å resolution. Another 3D class with twist 88.0, rise 44.2 Å was selected and refined to 3.9 Å, and postprocessing yielded the final map at 3.9 Å overall resolution.

The ADP-AlF$_4$-bound M854K MDA5 filament dataset was processed in the same way as the ADP-bound dataset except that the optics groups are not split.

The ATP-bound M854K MDA5 dataset was processed in the same way as the ADP-bound dataset except that dose-weighted images were used for CTF estimation. A reference-free two-dimensional (2D) class was generated from manual picking as a template for auto-picking. Helical reconstruction was performed using the ATP-bound WT MDA5-dsRNA structure (EMD-0024[14]) as the initial model. 1,068,108 auto-picked segments were subjected to three rounds of 2D classification yielding 383,128 particles. Three independent 3D classifications of ATP-bound filament segments did not show any significant differences in a twist and helical rise distribution of the selected segments. The ATP-bound segments were 3D-refined, CTF-refined, Bayesian-polished, and post-processed following the workflow shown in Supplementary Fig. 3e to yield the final 2.83 Å resolution map.

**Model building and refinement**. Previously reported cryoEM structures of MDA5-dsRNA filaments[14] were used as the starting atomic models: PDB:6G1X for the ADP-bound WT MDA5 structures; PDB:6GKH for the ADP-AlF$_4$-bound M854K MDA5 structures, and PDB:6GKM for the ATP-bound M854K MDA5 structure. The model was docked into the density for the central subunit in each map with the Fit in Map function in UCSF Chimera[40]. The docked model was rebuilt with COOT[41]. Models of adjacent protomers were generated in Chimera by applying the helical symmetry calculated in RELION. The resulting models with three MDA5 subunits were used for subsequent real-space refinement in PHENIX 1.18[42]. As described in ref. [14], global minimization and atomic displacement parameter (ADP) refinement options were selected in PHENIX. Secondary structure restraints, non-crystallographic symmetry restraints between the protein subunits, sidechain rotamer restraints, and Ramachandran restraints were used in real space refinement[14,42].

To determine which conformational state the helicase modules were in, each model was superimposed onto the fully closed structure of LGP2 (PDB:5JAJ[12]) using the secondary structure elements of Hel1 as the reference. The conformational state of the helicase domain was defined based on the rotation angles relating the Hel2 domains of the aligned structures as follows: closed state, 0°–3° angle; semi-closed state, 7°–13° angle[12].

**Statistics**. No statistical methods were used to predetermine sample size, experiments were not randomized, and the investigators were not blinded to experimental outcomes. Luciferase-reporter cell signaling data are represented as the mean ± standard error of the mean of three replicates conducted in a single independent experiment. Data were representative of at least three independent experiments. Scatter plots, histograms, and error bars were plotted with GraphPad Prism 9.2.0 and Microsoft Excel v.16.52.

**Reporting Summary**. Further information on research design is available in the Nature Research Reporting Summary linked to this article.

## Data availability
The data supporting the findings of this study are available from the corresponding authors upon reasonable request. The atomic coordinates were deposited in the Protein Data Bank (PDB) with accession codes 7BKP, 7BKQ, 7NGA, 7NIC, and 7NIQ. The cryoEM densities were deposited in the EM Data Bank with codes EMD-11937, EMD-

12092, EMD-12213, EMD-12288, and EMD-12294. The raw electron micrographs were deposited in EMPIAR with codes 10630, 10653, and 10664. Previously determined atomic coordinates used in this study are available in the PDB with codes 6G1X, 6GKH, 6GKM, 6H66, and 5JAJ. Source data are provided with this paper.

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

## Acknowledgements

We thank the following facility staff for assistance in cryoEM data collection: Bilal Ahsan, Giuseppe Cannone, Grigory Sharov, Shaoxia Chen, and other staff at the MRC-LMB EM Facility; Andrew Howe and Julika Radecke (eBIC); and Felix Weis (EMBL). We thank Kendra E. Leigh for assistance with negative-stain electron microscopy data collection. We thank Charles Rice (The Rockefeller University) for providing WT and ADAR1−/− HEK cells. We thank Jake Grimmett and Toby Darling (LMB Scientific Computing) for IT support for image processing. We thank Takanori Nakane, Paul Emsley, and Jude Short for helpful suggestions. We thank Brian Ferguson (Univ. of Cambridge) for comments on the manuscript and useful discussions. CryoEM data were collected at the MRC-LMB, eBIC at Diamond Light Source (DLS), and the European Molecular Biology Laboratory (EMBL). eBIC is supported by grant EM17434 from the Wellcome Trust, MRC, and BBSRC to DLS. This work was funded by Wellcome Trust Senior Research Fellowships 101908/Z/13/Z and 217191/Z/19/Z to Y.M.; Wellcome Trust Ph.D. Fellowship 215378/Z/19/Z to R.S.; and Human Frontier Science Program Long-Term Fellowship LT000454/2021-L to A.H.d.V. The work was supported by the NIHR Cambridge BRC (the views expressed are those of the authors and not necessarily those of the NIHR or the Department of Health and Social Care).

## Author contributions

Q.Y. and Y.M. conceived the experiments. Q.Y., A.H.d.V., and R.S. purified the proteins, performed the signaling assays, negative-stain electron microscopy, cryoEM data collection and analysis, and cryoEM image processing and reconstruction. Q.Y. performed the ATPase time-course assays. R.S. and A.H.d.V. performed the sample preparation and cryoEM reconstruction of ATP-bound M854K MDA5, the ATPase rescue assays, and signaling assays with short dsRNAs. All authors contributed to model building. Q.Y. refined the atomic models. All authors contributed to the figures. The first draft of the manuscript was written by Q.Y. and Y.M. and edited by all authors.

## Competing interests

Y.M. is a consultant for Related Sciences LLC and has profits interests in Danger Bio LLC. The remaining authors declare no competing interests.
