## [Peer Review File · Nature Communications]

REVIEWER COMMENTS

Reviewer #1 (Remarks to the Author):

In this manuscript Yu and Herrero Del Valle et al. perform an in-depth structure-function analysis to uncover the molecular basis for the auto-immunogenicity of the disease-related single point mutation M854K of the dsRNA specific pattern-recognition receptor MDA5. To this end they determine several high resolution cryoEM structures of dsRNA-MDA5 M854K filaments in the presence of different nucleotides and transition state analogues. In combination with biochemical characterisations as well as in cell signaling assays they are able to show that the MDA5 M854K mutant in all likelihood is activated by endogenous RNAs. M854K constitutively activates interferon signaling in cells albeit the isolated protein having an inactive ATPase activity. The M845K mutations forms additional polar contacts to the helicase domains in the dsRNA-MDA5 filament (in the ATP loaded state). This constrains the structure of the mutant filament in comparison with the wt filament and disrupts the ATP hydrolysis cycle. Since ATP hydrolysis is linked to proof-reading and dissociation from endogenous RNAs in wt MDA5, this explains why MDA5 M854K is constitutively active and auto-immunogenic.

All experiments are carefully designed and the present data are of impressive quality. The authors interpret the provided data carefully and more speculative interpretations are highlighted as such. In summary the manuscript and the data herein provide crucial information on an interesting aspect of immunology and RNA biology and will appeal to readers with interests as diverse as RNA and structural biology but also immunology as well as autoimmunity and infectious disease.

minor points:

-page 7, line 187: The authors state, that the MDA5 M854K protein precipitated after 4 to 7 days on ice which is in contrast to the wt MDA5 protein. They offer reduced stability as a likely reason. Can the authors exclude that the observed precipitation is caused by slow filament formation in the absence of ligand?

-page 8, Figure 2E: The negative stain data is fully consistent with many of the other data presented in this manuscript. However, a quantitation in addition to showing the exemplary micrographs of these data would strengthen further the conclusions drawn.

-page 11, Figure 3A, B and C: The authors might want to consider to 1) change the color of CARD domains (in (A)) vis-à-vis the RNA (in B and C), 2) highlight the ordered parts of MDA5 in the cartoon in (A) and 3) maybe highlight one MDA5 promoter in (B) and (C) for ease of understanding.

-page 12 line 298 to 306: The lower resolution and the broader twist distribution of the ADP-AIF4-M854K vs -wt filament reconstruction could potentially also be explained by the ~3-fold number of segments included in the former reconstruction. Have the authors attempted to classify the 167K ADP-AIF4-M854K class further to potentially obtain a more structurally homogenous population of ADP-AIF4-M854K segments?

-page 15, Figure 5(A): The authors might want to consider highlighting one MDA5 promoter in the filament again for ease of understanding.

-page 17, Figure 6: The authors might want to include their definition of "grip on RNA" in the figure legend (and how this is quantified). Do they mean increased stability of the filament, tighter packing of protomers or increased MDA5-dsRNA interaction interface area?

minor technical point:

-page 22, line 560/561: What is the reason for operating the K3 in CDS mode with the relatively elevated exposure/pix/s of ~20?

spelling:

- page 1, line 22, "of" missing: "CryoEM structures of MDA5-...."
- page 9, line 244, " b- and g-": should this maybe read " β - and γ -"?
- page 12, line 303, " b- and g-": should this maybe read " β - and γ -"?
- page 18, line 432, "of" missing: "with each of these.."
- page 18, line 438, " "with elevated interferon" : should this maybe read "with elevated interferon level/signaling and with the Singelton..."
- page 22, line 564, "Motion": should this maybe read "MotionCor2" or "with the MotionCor2 algorithm/routine/implementation"?

Reviewer #2 (Remarks to the Author):

Summary

The researchers performed extensive studies in understanding the mechanisms of a MDA5 mutant, M854K, found in autoinflammatory disorder AGS and SMS patients. They characterized the constitutive signaling activity of M854K mutant in the absence of exogenous RNA and the signaling response profile was distinct from gain- or loss-of-function mutations of MDA5 at the filament interface. The M854K MDA5 did not show ATPase activity while maintained affinity for RNA, even slightly higher than wild-type (WT) when binding to dsRNA. The lack of ATPase activity did not prevent the mutant from forming filaments. The authors also solved cryo-EM structures of nucleotide-bound M854K MDA5-dsRNA filament, in which M854K was found to form polar bonds with residues in the helicase domain, which constrained pincer and helicase domains compared to WT. The researchers hypothesized the constraints from the polar bonds by M854K mutation affected the MDA5 RNA footprint expansion and dissociation of ADP, which could prevent endogenous dsRNA dissociation, in accordance with M854K MDA5 constitutive signaling activity and lack of ATPase activity.

Strength

The authors started the M854K MDA5 investigation from two angles, filament formation and ATP binding/hydrolysis, which are the major factors in MDA5 malfunction. They narrowed down to the ATP hydrolysis since M854K mutant was able to form filaments. The high resolution cryo-EM structures revealed a clear picture of the key differences M854 MDA5 had compared to WT. The interactions M854K made with Hel domain restrained conformational change which could explain the RNA footprint expansion inhibition. The structures are well determined and beautifully presented.

Limitations

1. The proposed model of ATPase cycle based on cryo-EM dataset analysis is very speculative (Figure 6). The distribution of low, intermediate, and high as a function of steps in the ATP hydrolysis is interesting, however I would suggest that the authors minimize or remove the ADP release failure hypothesis. Other mechanisms remain possible, and more work is needed to clarify the ATPase inhibition step.
2. The malfunction of M854K MDA5 was probably from the constraints lysine side chain interacting with E813 and S491, it would be interesting to look if mutating E813 and S491 could free the constraints from M854K, and even restore function.
3. Line 115-117 and Figure 1 the author stated M854K MDA5 showed different signaling profile from mutations at the filament interface, but similar to mutations in the ATP or RNA binding sites. The disease mutations probably include R337G or R779C. Including these mutations in the signaling assays would solidify such statement.
4. Figure 1C, the author did protein expression titration experiment to assess MDA5 sensitivity and cooperativity. How was the M854K lacking selectivity concluded? Inducing WT and M854K MDA5 with short or Alu:Alu RNA may give more clear direction.
5. Line 138-158 seems not very close related to the main topic.
6. Line 151, "deltaC 12, had no signaling activity stimulation ...", there should be a "without" in

front of "stimulation".

7. Line 208 suggested a figure of WT MDA in the absence of dsRNA, which is missing.

8. Figure 5, labeling residue 822 would be helpful to Line 342.

9. Minor editorial issue – Line 303 "b" and "g" should be beta and gamma

Reviewer #3 (Remarks to the Author):

MDA5 is a major immune sensor for intracellular pathogenic RNA species. Unlike RIG-I, MDA5 prefers to bind long duplex RNA and form cooperative filaments for function. Given its unique role and RNA sensing preference, it could sense endogenous dsRNAs such as inverted Alu dsRNA. Several genetic autoimmune diseases have been associated to pathogenic MDA5 mutations. In this paper, the authors select a previous identified mutation M854K found in Aicardi-Goutières syndrome (AGS) and Singleton-Merten syndrome (SMS) patients to study. M854 located on the pincer helix that coordinates the helicase-ctd dsRNA binding domains but not directly interact with ATP or dsRNA binding. Using cell based MDA5 activation – IFN β transcription assay, the M854K mutant MDA5 seems to have increased basal and polyIC stimulated activities. Recombinant protein M854K abolishes the ATP hydrolysis activity but binds dsRNA more tightly than wt. The authors also determined three cryoEM structures of WT and M854K MDA5-dsRNA +/- ATP, ADP-AIF4, and ADP. Analysis of the structural differences suggest that M854K mutation keeps MDA5 in an activation-ready state by constrains the conformational flexibility that is necessary for ATP hydrolysis and its role in safe guard MDA5 from over-reacting to endogenous RNA species. Overall the data from different experiments largely agrees and supports the conclusion.

Major concerns:

- The missing cryoEM structure of M854K-dsRNA-ADP-Mg significantly weakens conclusion. Based on the authors' arguments and interpretation of the other structures, it seems that M854K-dsRNA-ADP-Mg complex would be a relatively more stable one. Could the authors obtain the structure and instead of hypothesizing its conformations / functional implications?
- Throughout the manuscript, there is an assumption "the M854K mutation enhances binding of MDA5 to endogenous RNAs to an extent that MDA5-dsRNA complexes are sufficiently stable to activate signalling". Could the author demonstrate this in the two cell lines used for the activity assay? Eg by cross-linking and pull-down experiment? Another experiment could be checking the M854K (vs wt) cellular response to dsRNAs of different lengths (100, 300, 500, etc).

Minor comments:

- Aicardi-Goutières syndrome (AGS) is inherited disease subacute encephalopathy caused by mutations in TREX1, RNASEH2B, RNASEH2C, RNASEH2A, SAMHD1, ADAR1, IFIH1. Rice GI et al, 2014, Nat Genet, proposed that mendelian disorders associated with an upregulation of type I interferon represent a distinct set of inborn errors of immunity related to mutations R337G, D393V, G495R, R720Q, R779V, R779C, R779H. Garau J et al, 2019, J Clin Med had performed whole-exome sequencing 51 AGS Italian patients found mutations on R337G, D393V, G495R, R720Q, R797H, R779C and M854K. Rice GI et al, 2014 demonstrated that mutations R337G, D393V, G495R, R720Q, R779V, R779C, R779H on MDA5 enhance binding affinity of MDA5 to dsRNA that confer gain of function of MDA5. Rice et al, 2014 also ruled out that these five mutants to assemble filaments more cooperatively than wild-type protein by cellular and biochemistry assay. How would the authors be sure that this M854K is unique and important in causing the disease?
- How about IFN- β signalling assay in ADAR1 over expression cell line? ADAR1 is an ISG, correct?
- Line 80, where is the reference MDA5 variant M854K associated SMS patient?
- Affinity of binding (Figure 2). Where is K_d value of the binding? Figure 2C and 2D are blurring and not clear and don't reflect affinity of the binding. Authors may refer the methods in publish of Rice GI et al 2014 to perform this assay and analyse data.
- Fig 3D, distances between K854 to S491 and E813?

We thank the three reviewers for their attentive reading of our manuscript and for their positive and constructive comments. We have carefully revised our manuscript and added new display items to address the reviewers' concerns. We have added a new supplementary figure (Supplementary Fig. 2), along with new main figure panels (Figs. 2d, 3g) and supplementary figure panels (Suppl. Fig. 1c). We have also revised the text and figures throughout to address each critique. Among these revisions, the most notable additions are:

- Addition of new ATP hydrolysis assays for M854K/S491A, M854K/E813A and M854K/S491A/E813A mutants of MDA5. The data demonstrate that Ser491 and Glu813 each play an important role in preventing M854K MDA5 from hydrolyzing ATP. We also show that ATPase activity is restored in the M854K/S491A/E813A triple mutant (Fig. 3g).
- Addition of new signaling assays with WT and M854K MDA5 using linear in vitro-transcribed dsRNA of different length as stimuli (instead of poly(I:C), see Supplementary Fig. 1c). These new data provide additional support for our conclusion that M854K MDA5 is less selective than WT and that the M854K variant is activated by endogenous RNAs in the absence of exogenous RNA stimuli.
- Addition of new negative-stain electron micrographs visualizing complex formation between M854K MDA5 but not WT MDA5 with linear 100-bp dsRNA (Fig. 2d). These new data provide additional support for our conclusion that M854K MDA5 forms more stable complexes with dsRNA than WT MDA5.
- Modification of our general mechanism of action of the M854K mutation to remove the hypothesis of inhibition of ADP release.

A copy of the manuscript with all substantive changes to the text highlighted has been attached as part of the review materials. Detailed point-by-point responses to the reviewers' comments follow below.

Point-by-point responses to reviewer comments

Reviewer #1 (Remarks to the Author):

In this manuscript Yu and Herrero Del Valle et al. perform an in-depth structure-function analysis to uncover the molecular basis for the auto-immunogenicity of the disease-related single point mutation M854K of the dsRNA specific pattern-recognition receptor MDA5. To this end they determine several high resolution cryoEM structures of dsRNA-MDA5 M854K filaments in the presence of different nucleotides and transition state analogues. In combination with biochemical characterisations as well as in cell signaling assays they are able to show that the MDA5 M854K mutant in all likelihood is activated by endogenous RNAs. M854K constitutively activates interferon signaling in cells albeit the isolated protein having an inactive ATPase activity. The M845K mutations forms additional polar contacts to the helicase domains in the dsRNA-MDA5 filament (in the ATP loaded state). This constrains the structure of the mutant filament in comparison with the wt filament and disrupts the ATP hydrolysis cycle. Since ATP hydrolysis is linked to proof-reading and dissociation from endogenous RNAs in wt MDA5, this explains why MDA5 M854K is constitutively active and auto-immunogenic.

All experiments are carefully designed and the present data are of impressive quality. The authors interpret the provided data carefully and more speculative

interpretations are highlighted as such. In summary the manuscript and the data herein provide crucial information on an interesting aspect of immunology and RNA biology and will appeal to readers with interests as diverse as RNA and structural biology but also immunology as well as autoimmunity and infectious disease.

We thank the reviewer for taking the time to carefully assess our manuscript. We are delighted they are so enthusiastic about our study.

Minor points:

-page 7, line 187: The authors state, that the MDA5 M854K protein precipitated after 4 to 7 days on ice which is in contrast to the wt MDA5 protein. They offer reduced stability as a likely reason. Can the authors exclude that the observed precipitation is caused by slow filament formation in the absence of ligand?

To address this question, we imaged a sample of MDA5 M854K in which precipitation had formed after several days of storage at 4°C by negative-stain electron microscopy. After precipitation had appeared, the sample was centrifuged 12,000 g for 10 min, loaded on an S200 Increase SEC column, and the void peak and the monomer peak were both collected and imaged separately. As can be seen in the images below, MDA5 M854K did not form any filaments in the absence of ligand. This is consistent with previous work by our group and others which has shown that MDA5 filament formation is strictly dependent on RNA binding. We agree with the reviewer, however, that it cannot be ruled out that other mutations that increase MDA5 signaling activity could in principle do so by forming filaments in the absence of ligand. We now also show in Fig. 2d a negatively stained EM image of (monomeric) MDA5 M854K without RNA in which no filaments are visible.

Aggregated M854K MDA5 (SEC void peak)

M854K MDA5 (SEC monomer peak)

-page 8, Figure 2E: The negative stain data is fully consistent with many of the other data presented in this manuscript. However, a quantitation in addition to showing the exemplary micrographs of these data would strengthen further the conclusions drawn.

Each micrograph in Fig. 2 with Alu:Alu dsRNA present is representative of 6 micrographs collected from three different meshes for each condition. In response to the reviewer's comment, we counted the number of filaments in each of the micrographs. We observed only two filaments in the images of WT MDA5 +Alu:Alu +ATP, versus 22 filaments for the M854K +Alu:Alu +ATP sample. No filaments were observed in the absence of Alu:Alu RNA. We have added a short statement in the text (p. 7) providing these numbers of filaments counted. We have also added a new supplementary figure (Supplementary Fig. 2 in the revised submission) showing all 12 micrographs in which Alu filaments were counted.

-page 11, Figure 3A, B and C: The authors might want to consider to 1) change the color of CARD domains (in (A)) vis-à-vis the RNA (in B and C), 2) highlight the ordered parts of MDA5 in the cartoon in (A) and 3) maybe highlight one MDA5 promoter in (B) and (C) for ease of understanding.

We thank the reviewer for these suggestions. We have changed the color of the CARDS to grey (not used elsewhere in the figure) and added horizontal bars in 3a to show which parts of the sequence are ordered in the cryo-EM structures. We have also outlined the central protomer in (b) and (c) as requested. The figure legend has been updated accordingly.

-page 12 line 298 to 306: The lower resolution and the broader twist distribution of the ADP-AIF4-M854K vs -wt filament reconstruction could potentially also be explained by the ~3-fold number of segments included in the former reconstruction. Have the authors attempted to classify the 167K ADP-AIF4-M854K class further to potentially obtain a more structurally homogenous population of ADP-AIF4-M854K segments?

We did originally attempt further classification for the entire ADP-AIF4-M854K particle population. As summarized in the workflow figure below (on the right-hand side), further classification did not lead to a more structurally homogenous population of ADP-AIF4-M854K segments with higher resolution. We note that the 167k and 62k ADP-AIF4-M854K

reconstructions have the same overall resolution (4.26Å) despite the 167k class having 2.7-fold more particles. The WT-AIF4 structure also had a similar resolution and narrower twist distribution than the 167k structure despite having only 3-fold fewer particles (as noted on p. 12 of the manuscript). Hence, we suspect that conformational heterogeneity in the 167k particle population that is limiting resolution, rather than particle numbers.

-page 15, Figure5(A): The authors might want to consider highlighting one MDA5 promoter in the filament again for ease of understanding.

The central MDA5 protomer has been highlighted as requested, as for Fig. 3.

-page 17, Figure6: The authors might want to include their definition of “grip on RNA” in the figure legend (and how this is quantified). Do they mean increased stability of the filament, tighter packing of protomers or increased MDA5-dsRNA interaction interface area?

We quantified the “grip on RNA” by measuring the MDA5-dsRNA interaction interface area with PISA. We have included below a table reporting the protein-RNA interface areas for eight selected structures (four from the present study and four previously reported). We would expect the protein-RNA interface area to correlate largely with stability of the filament, but we did not quantified filament stability.

Overall, the trend we observe is that MDA5 filaments with low helical twist (71-81°) have a greater protein-RNA interface area whereas filaments in high twist have smaller protein-RNA interfaces (see table below). Also, as we mention in the Discussion, increased protein-RNA interface area (tight grip) correlated with a narrower twist distribution, while relaxed grip correlated with a broader twist distribution, as quantified by twist distribution plots.

These trends are consistent with the higher resolution of the (low-twist) ATP- and AMPPNP-bound MDA5 filament reconstructions (because low-twist, ground-state filaments are structurally more ordered and have narrower twist distributions).

	Helical twist (low/interm./high)	Protein-RNA Interface area (Å ²)	N. of interface residues
854K-ATP	Low	2168	85
854K-AIF ₄ -62k	Interm.	2258	86
854K-AIF ₄ -167k	Interm.	2032	80
WT-ATP, PDB 6GKM	Low	2276	85
WT-ADP-AIF ₄ , PDB 6GKH	Interm.	2148	78
WT, PDB 6G1S	Interm.	2102	84
WT-ADP	High	2018	81
WT, PDB 6G1X	High	1925	75

We have added a mention of the protein-RNA interaction areas for the low-twist 854K-ATP and high-twist WT-ADP structures in the text (p. 9 and p. 14, respectively). We also mention in the Discussion that “MDA5 has a tight grip on the RNA, as evidenced by the larger protein-RNA interface...”. To avoid any confusion, we have also slightly modified the text on p. 16 and p. 18 to remove two instances of “grip” that were not based on the same protein-RNA contact area definition.

minor technical point:

-page22, line 560/561: What is the reason for operating the K3 in CDS mode with the relatively elevated exposure/pix/s of ~20?

This was an error in the text- we did not in fact use CDS mode. We have corrected this in the text (p. 22).

spelling:

- page 1, line 22, “of” missing: “CryoEM structures of MDA5-....”

Corrected.

- page 9, line 244, “ b- and g-“: should this maybe read “β- and γ-“?

Corrected.

- page 12, line 303, “ b- and g-“: should this maybe read “β- and γ-“?

Corrected.

- page 18, line 432, “of” missing: “with each of these..”

Corrected.

- page 18, line 438, “ “with elevated interferon” : should this maybe read “with elevated interferon level/signaling and with the Singelton...”

Corrected.

- page 22, line 564, “Motion”: should this maybe read “MotionCor2” or “with the MotionCor2 algorithm/routine/implementation”?

Corrected.

Reviewer #2 (Remarks to the Author):

Summary

The researchers performed extensive studies in understanding the mechanisms of a MDA5 mutant, M854K, found in autoinflammatory disorder AGS and SMS patients. They characterized the constitutive signaling activity of M854K mutant in the absence of exogenous RNA and the signaling response profile was distinct from gain- or loss-of-function mutations of MDA5 at the filament interface. The M854K MDA5 did not show ATPase activity while maintained affinity for RNA, even slightly higher than wild-type (WT) when binding to dsRNA. The lack of ATPase activity did not prevent the mutant from forming filaments. The authors also solved cryo-EM structures of nucleotide-bound M854K MDA5-dsRNA filament, in which M854K was found to form polar bonds with residues in the helicase domain, which constrained pincer and helicase domains compared to WT. The researchers hypothesized the constraints from the polar bonds by M854K mutation affected the MDA5 RNA footprint expansion and dissociation of ADP, which could prevent endogenous dsRNA dissociation, in accordance with M854K MDA5 constitutive signaling activity and lack of ATPase activity.

Strength

The authors started the M854K MDA5 investigation from two angles, filament formation and ATP binding/hydrolysis, which are the major factors in MDA5 malfunction. They narrowed down to the ATP hydrolysis since M854K mutant was able to form filaments. The high resolution cryo-EM structures revealed a clear picture of the key differences M854 MDA5 had compared to WT. The interactions M854K made with Hel domain restrained conformational change which could explain the RNA footprint expansion inhibition. The structures are well determined and beautifully presented.

We thank the reviewer for taking the time to review our work and for their constructive critique on our work.

Limitations

1. The proposed model of ATPase cycle based on cryo-EM dataset analysis is very speculative (Figure 6). The distribution of low, intermediate, and high as a function of steps in the ATP hydrolysis is interesting, however I would suggest that the authors minimize or

remove the ADP release failure hypothesis. Other mechanisms remain possible, and more work is needed to clarify the ATPase inhibition step.

We agree that mechanisms other than ADP release failure remain possible and that more work is needed to clarify in further detail how the ATPase activity is inhibited in the M854K mutant. In response to this critique, we have removed the ADP release failure hypothesis from the text.

Specifically, the following changes were made to the text:

- the last sentence in the Results section on p. 14 was rewritten and shortened, to remove mention of ADP dissociation.
- A sentence explicitly stating the ADP release hypothesis was deleted from the Results section (p. 16): *"We hypothesize that the M854K mutation may prevent ADP/Pi release by preventing motifs Vc-VI from becoming more conformationally dynamic in the post-hydrolysis state through the salt bridge between Lys854 and Glu813 in motif Vc."*
- The following sentence from the Discussion was deleted: *"Failure of ADP/Pi to dissociate would explain the observed lack of ATPase activity in the M854K variant."* (p. 18).
- A mention of *"ADP release"* on p. 18 (Discussion) was deleted.
- We deleted *"ADP/Pi release"* in Fig. 6 and we added a red question mark next to the red X on the ADP release step.

2. The malfunction of M854K MDA5 was probably from the constraints lysine side chain interacting with E813 and S491, it would be interesting to look if mutating E813 and S491 could free the constraints from M854K, and even restore function.

We thank the reviewer for this excellent suggestion. In response to this critique we expressed and purified three new MDA5 mutant proteins: M854K/S491A, M854K/E813A and M854K/S491A/E813A. We then measured their ATPase activities in comparison to (freshly purified) WT and M854K MDA5. We found that the double mutants (M854K/S491A, M854K/E813A) had no ATPase activity but the ATPase activity of the triple mutant M854K/S491A/E813A was restored to a comparable level to WT MDA5. Hence, we can conclude breaking both polar contacts with residues 813 and 491, respectively, restores most of the ATPase activity. Moreover, this new data confirms that both residues play an important role, as the presence of either single contact is sufficient to prevent ATP turnover. We have added this new ATP hydrolysis data as panel g in Fig. 3.

3. Line 115-117 and Figure 1 the author stated M854K MDA5 showed different signaling profile from mutations at the filament interface, but similar to mutations in the ATP or RNA binding sites. The disease mutations probably include R337G or R779C. Including these mutations in the signaling assays would solidify such statement.

Disease mutations R337G, R779C and R779H do indeed map to near the ATP site. Disease mutations D393V and G495R map to the RNA binding site. The signaling activities of these mutants have been reported previously by multiple groups, using the same type of cell-based luciferase reporter assay as we used in the present study. The IFN-beta signaling activities of R337G, R779C/H, D393V and G495R relative to WT and empty vector were reported in Ref. 26 (Rice *et al.* 2014, Fig. 3) and in Mandhana *et al.* 2018

(DOI:10.1089/jir.2018.0049, Fig. 1). Ref. 19 (Ahmad et al. 2018) also reports the signaling activities of R337G, R779H, D393V and G495R using a luciferase reporter assay (Fig. S1) along with a the IRF3 dimerization activity of R779H (Fig. S4). Rather than repeat these signaling assays, we have added a mention of R337G, R779C/H, D393V and G495R as specific examples of disease mutations mapping to the ATP and RNA binding sites (p. 4 and p. 7). We have also added citations to Ref. 19 and Mandhana et al. (new citation, from another group, Ref. 30), in addition to the existing citation to Ref. 26.

We note that R779C/H is listed in Ref. 26 as being “located near the ATP-binding site but... also in proximity to the protein-protein interface of the filament”. However, this statement was made before a detailed atomic model was available for the MDA5-dsRNA filament. CryoEM structures of the MDA5-dsRNA filament determined in 2018 (Ref. 14), showed that R779C/H did not overlap with the protein-protein filament interface.

4. Figure 1C, the author did protein expression titration experiment to assess MDA5 sensitivity and cooperativity. How was the M854K lacking selectivity concluded? Inducing WT and M854K MDA5 with short or Alu:Alu RNA may give more clear direction.

Firstly, the M854K variant showed significant activation in the absence of poly(I:C) RNA whereas WT MDA5 showed no signaling activation without poly(I:C) RNA. Moreover, the signaling from M854K without RNA was proportional to the expression level of M854K. Since no exogenous RNA was introduced in the experiment, the signaling activation observed with M854K must have been induced by cellular RNA (note that the EM images in Fig. 2d confirm that purified MDA5 M854K does not form filaments in the absence of RNA-see also responses to Reviewer 1 above). These observations suggest that M854K is less selective than WT against cellular RNA for signaling activation. This is supported by our observation that M854K MDA5 is more likely than WT MDA5 to form filaments on Alu:Alu dsRNA, which are abundant in the cell.

Secondly, when increasing the expression level of WT MDA5 with poly(I:C)-stimulation, we did not observe increased signaling proportional to WT MDA5 expression level. In contrast, upon increasing the expression level of M854K MDA5 with poly(I:C) stimulation, significantly increased signaling was observed. This suggests that WT MDA5 responds more cooperatively to RNA stimulation, whereas M854K MDA5 responds more proportionally.

We address the question of signaling induction by short (non Alu) dsRNAs in detail below See also new Supplementary Fig. 1c.

5. Line 138-158 seems not very close related to the main topic.

This passage compares the signaling activities of WT MDA5 with those of M854K and Δ C12 MDA5 variants. The passage includes descriptions of the selectivity and cooperativity of the signaling responses discussed above. We do think this topic will be of interest to readers but in response to this comment we have made a few edits to slightly shorten this paragraph.

6. Line 151, “deltaC 12, had no signaling activity stimulation ...”, there should be a “without” in front of “stimulation”.

Corrected- we thank the reviewer for spotting this.

7. Line 208 suggested a figure of WT MDA in the absence of dsRNA, which is missing.

We have added an image of WT MDA5 in the absence of dsRNA in Fig. 2, as requested. We have also added additional electron micrographs of WT and M854K MDA5 in the presence of Alu:Alu dsRNA, at the request of Reviewer 1 (Supplementary Fig. 2 in the revised submission).

8. Figure 5, labeling residue 822 would be helpful to Line 342.

Since only a small part of the side chain of residue 822 was visible in Fig. 5d and 5e, we have slightly modified the view in these two figure panels (clipping plane and translation in Z) to show more of the R822 side chain. We have added residue labels R822 in these two panels, as requested.

9. Minor editorial issue – Line 303 “b” and “g” should be beta and gamma

These typos have been corrected, thank you for spotting them.

Reviewer #3 (Remarks to the Author):

MDA5 is a major immune sensor for intracellular pathogenic RNA species. Unlike RIG-I, MDA5 prefers to bind long duplex RNA and form cooperative filaments for function. Given its unique role and RNA sensing preference, it could sense endogenous dsRNAs such as inverted Alu dsRNA. Several genetic autoimmune diseases have been associated to pathogenic MDA5 mutations. In this paper, the authors select a previous identified mutation M854K found in Aicardi-Goutières syndrome (AGS) and Singleton-Merten syndrome (SMS) patients to study. M854 located on the pincer helix that coordinates the helicase-ctd dsRNA binding domains but not directly interact with ATP or dsRNA binding. Using cell based MDA5 activation – IFN β transcription assay, the M854K mutant MDA5 seems to have increased basal and polyIC stimulated activities. Recombinant protein M854K abolishes the ATP hydrolysis activity but binds dsRNA more tightly than wt. The authors also determined three cryoEM structures of WT and M854K MDA5-dsRNA +/- ATP, ADP-AIF4, and ADP. Analysis of the structural differences suggest that M854K mutation keeps MDA5 in an activation-ready state by constrains the conformational flexibility that is necessary for ATP hydrolysis and its role in safe guard MDA5 from over-reacting to endogenous RNA species. Overall the data from different experiments largely agrees and supports the conclusion.

We thank the reviewer for taking the time to review our work and for their constructive critique on our work.

Major concerns:

- The missing cryoEM structure of M854K-dsRNA-ADP-Mg significantly weakens conclusion. Based on the authors' arguments and interpretation of the other structures, it seems that

M854K-dsRNA-ADP-Mg complex would be a relatively more stable one. Could the authors obtain the structure and instead of hypothesizing its conformations / functional implications?

In response to the request from Reviewer 2 to “minimize or remove the ADP release failure hypothesis” because it was too speculative and required follow up work, we have removed from the text the claim that ADP/Pi release is likely to be inhibited by the M854K mutation (see response to Reviewer 2 for a list of specific text changes). Therefore, in the revised submission, the lack of a structure of M854K-dsRNA-ADP-Mg will be much less obvious to the readership. We agree with the reviewer that such a structure would be interesting, but we feel that dissecting the mechanism of ATPase inhibition by the M854K mutation in further detail with structural studies is beyond the scope of the current study considering the revisions already made to the submission in response to other comments from all three reviewers.

- Throughout the manuscript, there is an assumption “the M854K mutation enhances binding of MDA5 to endogenous RNAs to an extent that MDA5-dsRNA complexes are sufficiently stable to activate signalling”.

We thank the reviewer for raising this point. We would like to clarify that we are not claiming that the M854K mutation necessarily enhances binding to endogenous RNAs but rather than it stabilizes MDA5-RNA complexes. This stabilization could be the result of enhanced binding but given the lack of ATPase activity in M854K MDA5 we think it is more likely that the principal source of stabilization is in fact a reduced dissociation rate. That is, binding affinities of M854K and WT could be similar, but M854K filaments would fail to disassemble (or disassemble more slowly) in the presence of ATP. We found two instances in the text where the M854K variant is said to “enhance binding to dsRNA” (on p. 3, end of Introduction and at the top of p. 7, in Results). We have rephrased both passages in terms of RNA complex stabilization. We also changed “M854K MDA5... binds more tightly to... dsRNA” to “M854K MDA5... binds more stably to... dsRNA” in the abstract.

Could the author demonstrate this in the two cell lines used for the activity assay? Eg by cross-linking and pull-down experiment?

We agree that our conclusion that MDA5 stabilizes complex formation with endogenous dsRNA ligands is somewhat indirect. However, the disease association of the M854K allele, and the strong IFN response with M854K in our cellular assay in the absence of exogenous RNA stimuli, support that model that the M854K mutation stabilizes MDA5-RNA complexes with endogenous RNAs to an extent sufficient to produce significant IFN signaling.

Protein-RNA complexes are likely to be too unstable to provide more direct supporting evidence using pulldown experiments, and crosslinking experiments produce high background signal from non-specific crosslinking in our experience. Instead, we have performed the alternative new experiment proposed by the reviewer in the next point: assessing the cellular response to dsRNAs of different lengths.

Another experiment could be checking the M854K (vs wt) cellular response to dsRNAs of different lengths (100, 300, 500, etc).

We thank the reviewer for this suggestion. In response to this comment, we performed a new set of cell signaling experiments in which we measured the IFN luciferase reporter responses of WT and M854K MDA5 with different types of RNA ligands, including 100-bp, 300-bp and 1-kbp dsRNA. We synthesized the RNAs by in vitro transcription and purified them using silica-based columns. We included an empty vector control. The results are included in a new figure panel in Supplementary Figure 1 (panel c in the revised submission) and summarized on p.5. For the clearest visualization of the effect of using different RNA ligands, we compared the cellular responses with the dsRNA ligand panel to the response with polyI:C for each MDA5 variant. We found that the luciferase reporter response in cells stimulated with 100-bp, 300-bp and 1-kbp dsRNA were similar as in cells stimulated with polyI:C for M854K MDA5. In contrast, for WT MDA5 the cellular response was significantly smaller with all three in vitro-transcribed RNAs than with polyI:C. Although we might have expected the response with the 1-kbp RNA to be more similar to polyI:C for WT MDA5, we attribute the lower-than-expected response to less efficient transfection of linear transcribed dsRNAs versus polyI:C, in which extensive crosslinking is known to promote cellular uptake. Overall, these new results support our original conclusion on p. 5 that M854K MDA5 is less selective than WT and in the cellular context M854K MDA5 is activated by endogenous RNAs even in the absence of stimulation with exogenous RNA ligands.

To complement this new cell signaling data we also imaged purified recombinant WT and M854K MDA5 in the presence of 100-bp dsRNA by negative-strain electron microscopy, to assay for MDA5-RNA filament formation. We found that M854K MDA5 formed clear filaments on 100-bp dsRNA, whereas no filaments were visible with WT MDA5. Representative images of M854 and WT MDA5 with 100-bp dsRNA are shown in Fig. 2d and summarized on p. 7.

Minor comments:

- Aicardi-Goutières syndrome (AGS) is inherited disease subacute encephalopathy caused by mutations in TREX1, RNASEH2B, RNASEH2C, RNASEH2A, SAMHD1, ADAR1, IFIH1. Rice GI et al, 2014, Nat Genet, proposed that mendelian disorders associated with an upregulation of type I interferon represent a distinct set of inborn errors of immunity related to mutations R337G, D393V, G495R, R720Q, R779V, R779C, R779H. Garau J et al, 2019, J Clin Med had performed whole-exome sequencing 51 AGS Italian patients found mutations on R337G, D393V, G495R, R720Q, R797H, R779C and M854K. Rice GI et al, 2014 demonstrated that mutations R337G, D393V, G495R, R720Q, R779V, R779C, R779H on MDA5 enhance binding affinity of MDA5 to dsRNA that confer gain of function of MDA5. Rice et al, 2014 also ruled out that these five mutants to assemble filaments more cooperatively than wild-type protein by cellular and biochemistry assay. How would the authors be sure that this M854K is unique and important in causing the disease?

The genetic association of the M854K with interferonopathies is strong because the M854K allele (*IFIH1* 2561T>A) was detected in in patients from unrelated families in Italy and Japan. The association in the two clinical studies we cite (Refs. 28, 29) is clear, involves a similar set

of clinical symptoms, and the M854K allele did not appear to be linked with any other alleles so we can be confident that M854K is causal and associated with disease.

Regarding the mechanism of action in relation to previously reported variants, the key difference between M854K and all other disease variants described previously is that M854K does not map to the ATP binding site or dsRNA binding interface. All mutants listed above either inhibit nucleotide binding (eg. R337G) or enhance dsRNA binding (eg R779V/C/H), whereas we show that M854K interferes with conformational changes coupled to ATP hydrolysis. Hence the M854K variant is unique in that it allosterically interferes with conformational changes required for RNA proofreading by MDA5.

A further unique property of M854K that it responds to RNA less cooperatively than WT, whereas the mutants described in Rice et al (2014) are fact described by the authors to bind RNA more cooperatively, contrary to the reviewer's interpretation. Indeed, the statement in Rice et al (2014) regarding the cooperativity of the disease mutants is that the EMSA data "rules out a lack of ATP hydrolysis as a reason for these five mutants to assemble filaments more cooperatively than wild-type protein" (p. 508, first paragraph). The previous two sentences state that "the population of intermediate-size complexes was markedly diminished with all six mutants..." and "...with the exception of R337G, all the mutants hydrolyzed ATP as well as wild-type protein". Together, the implication of these statements is that all six disease mutants bind RNA more cooperatively than WT – as evidenced by the by smaller population of intermediate-size complexes – and that loss of ATPase activity can be ruled out as the cause for this increased cooperativity for five of the mutants (all except R337G, which is ATPase-dead).

- How about IFN- β signalling assay in ADAR1 over expression cell line? ADAR1 is an ISG, correct?

This is a good suggestion. We expect IFN- β signaling to be similar in cell overexpressing ADAR1 and in WT cells, however, we do not feel the data from this experiment would be sufficiently physiologically relevant or novel to justify resourcing it, given that we do not have an ADAR1 overexpression cell line in our laboratory. Moreover, variability in the ADAR1 expression level in overexpressing cells is likely to make it challenging to compare experiments with one another.

- Line 80, where is the reference MDA5 variant M854K associated SMS patient?

Two of the cited references, Refs. 28 and 29, report an associated of the M854K variant with SMS in human patients.

- Affinity of binding (Figure 2). Where is K_d value of the binding? Figure 2C and 2D are blurring and not clear and don't reflect affinity of the binding. Authors may refer the methods in publish of Rice GI et al 2014 to perform this assay and analyse data.

Gel-shift assays (EMSA) are unsuitable for accurate measurement of the K_d. This is because gel electrophoresis is not an equilibrium experiment and hence cannot be used to accurately measure equilibrium constants, including dissociation constants. We are aware

that it is relatively common in the literature for an apparent K_d to be calculated from EMSAs by gel densitometry, for example as in the reference cited by the reviewer. Although an approximate K_d can be estimated from an EMSA we would argue that providing detailed K_d calculation with the EMSA would be potentially misleading as the resulting K_d value is bound to be unreliable. Indeed, K_d s estimated from EMSAs usually underestimate the binding affinity as electrophoresis tends to prevent reassociation of any dissociated complexes.

Despite these limitations we agree that the K_d can generally still be estimated to within an order of magnitude as the protein concentration at which half of the RNA is shifted away from the unbound RNA band. Using this reasoning, 100 nM is within an order of magnitude of the K_d for the EMSAs shown in Fig. 2 (and the actual K_d values at equilibrium are likely to be lower). Despite the blurriness of the gel bands, it is clearly apparent in panel b that M854K MDA5 binds Alu:Alu dsRNA more tightly than WT MDA5, and in panel c that Alu+ssRNA binds more weakly to both WT and the M854K variant than Alu:Alu dsRNA. However, we feel it would be misleading to attempt to quantify the exact K_d s for each experiment given the limitations of EMSAs in general for K_d determination.

- Fig 3D, distances between K854 to S491 and E813?

The distance between the residues has been added to Fig. 3 (panel e in the revised submission): K854(NZ)-E813(OE2): 2.74; K854(NZ)-E813(OE1): 3.74Å; K854(NZ)-S813(OG): 2.81 Å.

REVIEWERS' COMMENTS

Reviewer #2 (Remarks to the Author):

The authors have done an excellent job at addressing my concerns.

Reviewer #3 (Remarks to the Author):

The authors have addressed my comments well and revised the manuscript accordingly. No further comments.